# TALK LIKE A GRAPH: ENCODING GRAPHS FOR LARGE LANGUAGE MODELS

**Bahare Fatemi, Jonathan Halcrow, Bryan Perozzi**
Google Research
{baharef,halcrow,bperozzi}@google.com

## ABSTRACT

Graphs are a powerful tool for representing and analyzing complex relationships in real-world applications such as social networks, recommender systems, and computational finance. Reasoning on graphs is essential for drawing inferences about the relationships between entities in a complex system, and to identify hidden patterns and trends. Despite the remarkable progress in automated reasoning with natural text, reasoning on graphs with large language models (LLMs) remains an understudied problem. In this work, we perform the first comprehensive study of encoding graph-structured data as text for consumption by LLMs. We show that LLM performance on graph reasoning tasks varies on three fundamental levels: (1) the graph encoding method, (2) the nature of the graph task itself, and (3) interestingly, the very structure of the graph considered. These novel results provide valuable insight on strategies for encoding graphs as text. Using these insights we illustrate how the correct choice of encoders can boost performance on graph reasoning tasks inside LLMs by 4.8% to 61.8%, depending on the task.

## 1 INTRODUCTION

There has been remarkable recent progress in the research and applications of large language models (LLMs) (Vaswani et al., 2017; Devlin et al., 2018; Brown et al., 2020a; Ouyang et al., 2022). These generative models have captivated the artificial intelligence community and a plethora of models trained on a variety of tasks and modalities have recently been released (Zhao et al., 2023). All of these advancements have led to a growing consensus that LLMs are a pivotal advancement on the path to artificial general intelligence (AGI) (Bubeck et al., 2023).

However, despite all their successes, there are a number of limitations with the current methodology of design and implementation of LLMs. One of the most obvious limitations is their reliance on unstructured text, causing the models to sometimes miss obvious logical entailments or *hallucinate* incorrect conclusions (Zhang et al., 2023b). Another is that LLMs are fundamentally limited by when they were trained, and it can be difficult to incorporate 'fresh' information about the state of the world which has changed (Lewis et al., 2020). Graph-structured data is one of the most flexible ways to represent information and could be a promising solution to both challenges (Schneider et al., 2022; Pan et al., 2023). Graphs can provide LLMs with a more explicit representation of relationships and entities, enabling them to reason more effectively and avoid hallucinations. Graphs can be updated and expanded, allowing LLMs to incorporate new information as it becomes available.

Interestingly, despite this promise, the intersection of graphs and LLMs has been relatively understudied. For example, while much work has focused on LLMs and graph databases (or *knowledge graphs* (Guu et al., 2020; Lewis et al., 2020)) there has not been much study about general purpose use of graph-structured data. More recently, Wang et al. (2023) have sought to address this by designing a graph benchmarking task for language models. While their task represents an exciting initial foray into measuring LLMs graph reasoning capabilities, there are still many open questions due to the omission of several natural graph tasks and a lack of variety in the type of graph structure considered. Other recent work seeks to replace graph-structured data with LLMs (Ye et al., 2023), but this does not address fundamental challenges with LLMs.

In this work, we perform the first comprehensive study about reasoning over graph-structured data as text for consumption by LLMs. To analyze graph reasoning more closely, we decompose the

Figure 1: Overview of our framework for reasoning with graphs using LLMs.

problem into *graph encoding* and *graph prompt engineering*. Varying graph encoding methods allows us to understand how LLM's learned representations are leveraged in graph tasks. While studying prompt engineering techniques finds the most suitable way to get a desired solution to a question from an LLM. Our experimental results seek to uncover the situations where different prompt heuristics work well. To that end, we propose a new set of benchmarks `GraphQA` for measuring LLM performance reasoning over graph data. `GraphQA` is distinguished by using graphs with much more varied and realistic graph structure than has previously been studied with LLMs[1].

**Our Contributions**: Specifically, the contributions of our work are the following:

1. An **extensive study** of graph-structure prompting techniques for use in LLMs.
2. **Insights and best practices** for encoding graphs as text for use in LLMs.
3. A new **graph benchmark** (`GraphQA`) to aid the community in studying the effects of graph structure on LLM prompting further.

## 2 PROMPTING LLMS FOR GRAPH REASONING

**Notation.** Let $f$ be the interface function to a generative AI model, which takes high-dimensional discrete input tokens $W$ and produces output in the same token space ($f : W \mapsto W$). Without loss of generality, we will colloquially refer to $f$ as a pre-trained Large Language Model (LLM) throughout this work, but note that our discussion here applies to any generative AI model with such a discrete interface. In this work, we consider encoding graphs $G = (V, E)$, where $V$ is the set of vertices (or nodes) and $E \in (V \times V)$ is the set of edges connecting them.

### 2.1 PROMPT ENGINEERING

The goal in prompt engineering is to find the correct way to phrase a question $Q$ such that an LLM $f$ (or other generative model) will return the corresponding answer $A$, ($Q \in W, A \in W$). In other words, $A = f(Q)$. In this work, our goal is to provide the LLM $f$ with graph information $G$, so that it can better reason about question/answer pairs that require access to arbitrarily structured relational information: $A = f(G, Q)$.

A variety of approaches exist for modifying the LLM $f(.)$ so that it could perform better on tasks with graph data such as fine-tuning (Clark et al., 2020), soft prompting (Lester et al., 2021), and LoRA (Hu et al., 2021). In addition, many approaches modify the model to include graph information (Müller et al., 2023; Zhang et al., 2020; Dwivedi & Bresson, 2020). However, these methods require access to the internals of the model (its weights or gradients), which can limit their applicability in many settings. In this work, we are instead interested in the case where $f(.)$ and its parameters are fixed, and the system is available only for use in a *black box* setup where the LLM only consumes and produces text (*i.e.*, the LLM $f : W \mapsto W$). We believe this setting to be particularly valuable as the number of proprietary models available and their hardware demands increase.

To this end, we introduce the graph encoding function $g(G)$ and question rephrasing function $q(Q)$, where $g : G \mapsto W$ and $q : W \mapsto W$ (where $W$ is the large discrete domain of tokens used to train the LLM) as $A = f(g(G), Q)$. Our training input $D$ to the graph-based prompt system is a set of $G, Q, S$ triples, where $G$ is a graph, $Q$ is a question asked to the LLM, and $S$ is a solution to $Q$, ($S \in W$). We seek to find a $g(.)$ and $Q$ that maximize the expected score from the model (score$_f$) of the answers over the training dataset $D$.

$$\max_{g,q} \mathbb{E}_{G,Q,S \in D} \text{score}_f(g(G), q(Q), S) \tag{1}$$

---

[1] The code to generate the data is available at `https://github.com/google-research/google-research/tree/master/graphqa`.

As $W$ is a very large discrete space, many current approaches use heuristics for this optimization (by changing the prompt $Q$). The novel contribution of this work is to consider the role of the graph encoding function $g(.)$, question rephrasing function $q(.)$, and the graph structure $G$ in the optimization of Equation (1).

## 2.2 PROMPTING HEURISTICS

The vast majority of prompting heuristics operate by optimizing the prompt text $Q$ used to query the model. We briefly introduce the methods we examine further in the paper here:

**Zero-shot prompting** (ZERO-SHOT): This approach simply provides the model with a task description and asks it to generate the desired output, without any prior training on the task. **Few-shot in-context learning** (FEW-SHOT) (Brown et al., 2020b): This approach provides the model with a small number of examples of the task, along with the desired outputs. The model then learns from these examples to perform the task on new inputs. **Chain-of-thought** (CoT) prompting (COT) (Wei et al., 2022): This approach provides the model with a sequence of examples, each of which shows how to solve the task step-by-step. The model then learns to generate its CoTs to solve new problems. **Zero-shot CoT** prompting (ZERO-COT) (Kojima et al., 2022): This approach is similar to CoT prompting, but it does not require any prior training examples. Instead, the model uses a simple prompt to generate its own CoTs. As suggested by the original paper, we used "Let's think step by step". **Bag prompting** (COT-BAG) (Wang et al., 2023): This technique is proposed to improve the performance of LLMs on graph-related tasks. It works by appending "Let's construct a graph with the nodes and edges first" to the graph description.

We note that there is also a popular recent extension of this family of methods, based on *iterative prompting*. These methods use a series of iterative LLM queries to optimize the prompt question (*e.g.*, (Zhou et al., 2022b; Pryzant et al., 2023; Yang et al., 2023)). However, our initial experiments showed that iterative prompting methods performed much worse for our tasks, due to cascading errors. Consequently, we chose to concentrate our efforts on the methods outlined above. In this study, the goal is to optimize the graph encoding function on basic graph tasks. Such basic tasks are essential intermediate steps for more complex reasoning tasks on graphs. We conduct extensive experiments on graph encoding function, question, and graph generator functions, providing a study of graph encoding methods for black-box LLM usage.

## 3 TALK LIKE A GRAPH: ENCODING GRAPHS VIA TEXT

Graph encoding is a necessary step for turning graph-structured information into a sequence for consumption by language models. In this section, we will study the details of a graph encoding function $g(.)$ which maps graph data into tokens for consumption by an LLM. Our experimental results in this section seek to understand the best form of graph encoding and prompt engineering to maximize the performance on graph reasoning tasks.

We begin by highlighting some of the most exciting results from our analysis here:

- **R1**: LLMs perform poorly on basic graph tasks (§3.1).
- **R2**: The graph encoding function has a significant impact on LLM graph reasoning (§3.1).
- **R3**: Incident graph encoding outperforms the rest in most of the setups (§3.1).
- **R4**: Model capacity has a significant effect on graph reasoning capabilities of LLMs (§3.4).

**Graph encoding function**. This section is an investigation into various methodologies for representing graphs as text. This process of encoding graphs as text can be separated into two key inquiries: First, the encoding of nodes in the graph, and second the encoding of edges between the nodes. Regarding the encoding of nodes and edges, we examine several techniques. Figure 2 shows an overview of the graph encoding functions used. For brevity's sake, a full description and examples of the graph encoding functions considered are explained in Appendix A.1.

**Graph structure**. We briefly note that the design of this experiment follows that of Wang et al. (2023), who use Erdős-Rényi (ER) graphs (Erdős & Rényi, 1959). One contribution of our work is to consider the effect of more complex graph structures on reasoning in LLMs (covered in Section 4).

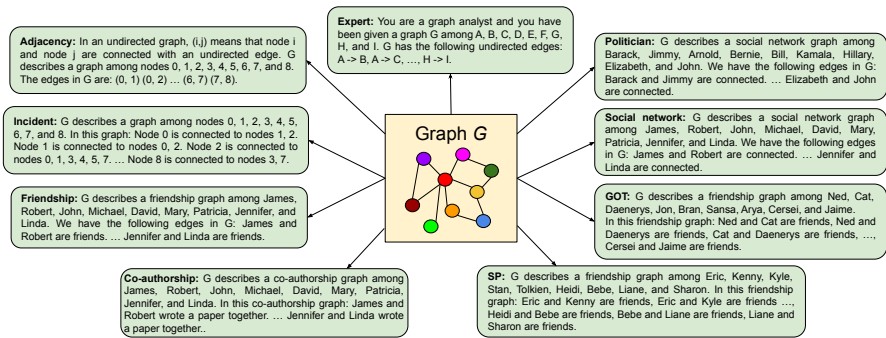

Figure 2: Overview of our framework for encoding graphs via text.

## 3.1 EXPERIMENT 1: VARYING GRAPH ENCODING FUNCTION

In this experiment, we measure the performance of pre-trained LLMs on graph tasks: *edge existence*, *node degree*, *node count*, *edge count*, *connected nodes*, and *cycle check*. We describe these tasks and our graph benchmark that contains them (GraphQA) in detail in Appendix A.2.

### 3.1.1 RESULTS

Table 1 shows the results of this experiment varying graph encoding and prompting techniques. These results show several interesting conclusions, which we briefly summarize here:

**LLMs perform poorly on basic graph tasks.** Let's start by examining the overall results. LLMs performed poorly on almost all the basic graph tasks we experimented with. This is especially interesting for *edge existence* and *cycle check*, where there is not an edge 53.96% of the time for *edge existence* and there is a cycle 81.96% of the time for *cycle check*. Therefore. LLMs perform worse than the majority baseline. Note that we experimented with ER graphs in this experiment, and it is very likely for an ER graph to have a cycle.

**Simple prompts are best for simple tasks.** We see that ZERO-COT prompting has worse model performance than ZERO-SHOT prompting on basic graph tasks. This is likely because ZERO-SHOT prompting is sufficient for these tasks, which do not require multi-hop reasoning. ZERO-COT prompting can be effective for tasks that require multi-hop reasoning, such as arithmetic problems, but it is not necessary for most basic graph tasks, which only require the LLM to have an understanding of the graph structure (nodes, edges, paths, etc.) and the graph task. However for more complex tasks, adding few-shot examples and CoT prompting generally improved the performance of the model. This is mainly because few-shot examples provide the LLM with a better understanding of the task it is solving. CoT prompting can also improve performance by helping the LLM to find out how to get to the answer to the problem.

**Graph encoding function has significant impact on LLM reasoning.** As the results indicate, the choice of the graph encoding function has a significant impact on the performance of LLMs on graph-related tasks. This is because different encoding functions capture different aspects of the graph structure. For instance, for finding connected nodes to a node in a graph, adjacency achieves 19.8% accuracy and incident achieves 53.8% accuracy. For both node degree and connected nodes, incident encoding outperforms the rest of the encoding functions. This is likely because the incident encoding encodes the graph structure in a way that makes the relevant information more accessible, *i.e.*, in close proximity, to the LLM.

**Integer node encoding improves arithmetic performance.** Another finding here is that integer encoding of nodes (*e.g.*, *node 0*) can improve the performance of LLMs on integer output tasks, such as predicting node degree, node count, and edge count. This is because the input and output of the LLM are then in the same space, making it easier for the model to learn the relationship between the two. Interestingly however, encoder functions with specific names (*e.g.*, David) worked better in non-integer output tasks such as GOT for *edge existence* or Friendship for *cycle check*.

**Aggregated results.** To provide recommendations about the best encoding function for each prompt, we rank the encoders by their average standing (in rank order) on each graph task. For most prompting methods, incident encoding performed the best. However, for ZERO-SHOT graph prompting,

Table 1: Comparison of various graph encoding functions based on their accuracy on different graph tasks using PaLM 62B. The most effective prompting heuristic is highlighted with an underline, and the top-performing graph encoding function for it is highlighted in bold. The overall result is represented its average ($\mu$) and an absolute difference ($\delta$) of its best and worst graph encoding.

| Method | Encoding | Edge existence | Node degree | Node count | Edge count | Connected nodes | Cycle check |
|---|---|---|---|---|---|---|---|
| ZERO-SHOT | Overall ($\mu/\delta$) | 44.5 / 9.4 | 14.0/16.0 | 21.73 / 8.6 | 12.4 / 4.8 | 14.7 / 11.0 | 76.0 / 13.2 |
| | Adjacency | 45.8 | 12.4 | 18.8 | 14.0 | 19.8 | 71.6 |
| | Incident | 39.6 | 25.0 | 15.6 | 10.6 | 53.8 | 68.8 |
| | Co-authorship | 44.0 | 13.8 | 22.0 | 11.4 | 7.6 | 70.8 |
| | Friendship | 46.6 | 11.2 | 23.0 | 10.2 | 4.0 | **82.0** |
| | SP | 46.4 | 9.0 | 22.4 | 15.0 | 6.2 | 80.4 |
| | GOT | **49.0** | 13.6 | 22.8 | 13.2 | 7.6 | 79.0 |
| | Social network | 43.2 | 16.0 | 22.8 | 10.8 | 8.2 | 81.2 |
| | Politician | 44.6 | 15.2 | 24.2 | 11.6 | 8.8 | 81.0 |
| | Expert | 41.2 | 10.0 | 24.0 | 14.8 | 16.4 | 69.6 |
| ZERO-COT | Overall ($\mu/\delta$) | 33.5 / 11.6 | 10.4 / 22.4 | 14.6 / 9.4 | 9.4 / 4.8 | 8.8 / 9.2 | 32.3 / 23.2 |
| | Adjacency | 34.2 | 15.4 | 11.0 | 12.2 | 6.0 | 46.2 |
| | Incident | 41.4 | 26.6 | 10.0 | 12.2 | 35.2 | 39.0 |
| | Co-authorship | 29.8 | 9.8 | 15.6 | 8.2 | 3.0 | 28.2 |
| | Friendship | 28.4 | 7.0 | 19.4 | 7.4 | 3.0 | 31.2 |
| | SP | 32.6 | 9.2 | 15.6 | 8.4 | 5.0 | 34.8 |
| | GOT | 34.6 | 8.4 | 16.2 | 8.4 | 5.4 | 33.4 |
| | Social network | 30.8 | 6.6 | 14.0 | 9.2 | 3.8 | 26.0 |
| | Politician | 38.0 | 4.2 | 14.6 | 8.6 | 3.2 | 23.0 |
| | Expert | 31.6 | 6.0 | 14.8 | 10.0 | 14.2 | 28.8 |
| FEW-SHOT | Overall ($\mu/\delta$) | 36.8 / 13.8 | 17.4 / 23.4 | 25.3 / 35.6 | 12.0 / 9.0 | 12.4 / 15.2 | 37.4 / 24.0 |
| | Adjacency | 42.8 | 15.4 | 47.2 | 18.6 | 22.2 | 47.8 |
| | Incident | 38.8 | 33.6 | 51.2 | 14.6 | 36.6 | 45.0 |
| | Co-authorship | 29.4 | 15.6 | 15.6 | 10.2 | 9.0 | 46.8 |
| | Friendship | 40.6 | 12.2 | 18.4 | 9.8 | 6.4 | 41.4 |
| | SP | 34.6 | 18.0 | 18.0 | 12.0 | 6.8 | 38.2 |
| | GOT | 40.6 | 17.2 | 14.2 | 12.0 | 3.4 | 28.6 |
| | Social network | 37.4 | 15.0 | 21.2 | 10.2 | 7.8 | 34.2 |
| | Politician | 38.0 | 13.4 | 21.4 | 9.6 | 7.8 | 30.8 |
| | Expert | 29.0 | 16.6 | 20.4 | 11.2 | 11.8 | 23.8 |
| COT | Overall ($\mu/\delta$) | 42.8 / 7.0 | 29.2 / 60.4 | 27.6 / 42.4 | 12.8 / 17.4 | 13.1 / 18.0 | 58.0 / 16.4 |
| | Adjacency | 42.8 | 71.2 | 57.0 | **25.2** | 22.4 | 56.6 |
| | Incident | 41.6 | **75.0** | **57.6** | 21.4 | 30.2 | 62.6 |
| | Co-authorship | 43.2 | 16.4 | 15.2 | 8.8 | 8.4 | 54.8 |
| | Friendship | 46.6 | 14.6 | 23.0 | 7.8 | 9.6 | 61.8 |
| | SP | 42.6 | 17.4 | 17.0 | 10.6 | 8.2 | 59.4 |
| | GOT | 44.0 | 17.8 | 16.2 | 11.8 | 7.2 | 60.4 |
| | Social network | 42.6 | 16.4 | 21.6 | 8.4 | 8.0 | 60.6 |
| | Politician | 42.2 | 16.6 | 22.6 | 9.2 | 9.4 | 59.4 |
| | Expert | 39.6 | 17.4 | 18.0 | 12.4 | 14.4 | 46.2 |
| COT-BAG | Overall ($\mu/\delta$) | 37.3 / 16.6 | 28.0 / 61.8 | 26.9 / 33.8 | 12.5 / 17.8 | 15.8 / 31.8 | 52.1 / 26.0 |
| | Adjacency | 45.8 | 66.8 | 48.6 | 25.0 | 20.6 | 56.8 |
| | Incident | 45.6 | 75.2 | 51.2 | 21.8 | **41.0** | 63.0 |
| | Co-authorship | 25.0 | 14.6 | 17.4 | 7.2 | 9.2 | 37.0 |
| | Friendship | 39.0 | 16.2 | 21.8 | 7.4 | 9.8 | 52.0 |
| | SP | 33.6 | 17.0 | 21.6 | 11.4 | 11.4 | 52.2 |
| | GOT | 32.6 | 15.6 | 18.0 | 11.0 | 10.0 | 54.6 |
| | Social network | 44.8 | 13.4 | 19.6 | 9.0 | 10.0 | 51.2 |
| | Politician | 40.4 | 17.6 | 22.8 | 8.2 | 10.2 | 57.2 |
| | Expert | 29.2 | 15.8 | 20.8 | 11.6 | 20.4 | 45.0 |

node tokens with more established representations (such as politicians) outperformed incident. The results are deferred to Table 6 in the appendix.

**Summary:** Choosing the right graph encoding function significantly affects the performance of LLMs on graph algorithms. Therefore, it is important to select a function carefully and appropriately for the specific task. This finding is especially important because many reasoning tasks involve graph problems. For example, finding influential nodes in a social network is similar to finding the degree of the nodes in the graph. Encoding such graphs in the right way for the task can improve the task.

## 3.2 EXPERIMENT 2: VARYING PROMPT QUESTIONS

The motivation for this experiment is to measure the effect of the question encoder in the prompt on different graph tasks. In this experiment, we maintained the graph encoding function as a constant

Table 2: Comparing two question encoders based on their accuracy for PaLM 2 XXS and PaLM 62B. The top-performing question encoder for the respective LLM is highlighted in bold.

| Method | Question encoder | LLM | Edge Existence | Node degree | Node count | Edge count | Connected nodes |
|---|---|---|---|---|---|---|---|
| ZERO-SHOT | Graph | PaLM 2-XXS | 42.8 | 10.8 | 5.4 | **5.6** | 1.6 |
| | Application | PaLM 2-XXS | **60.8** | **14.0** | **9.4** | 4.4 | **11.4** |
| | Graph | PaLM 62B | 46.6 | 11.2 | **23.0** | 10.2 | 4.0 |
| | Application | PaLM 62B | **47.8** | **16.6** | 17.8 | **13.2** | **6.0** |
| COT | Graph | PaLM2 XXS | 50.4 | 8.8 | 8.4 | 4.2 | 10.2 |
| | Application | PaLM2 XXS | **56.4** | **12.2** | **8.6** | **5.4** | **11.0** |
| | Graph | PaLM 62B | **46.6** | 14.6 | **23.0** | 7.8 | 9.6 |
| | Application | PaLM 62B | 38.6 | **16.6** | 16.0 | **12.2** | **10.0** |

| | Task | Same relation | Multiple relations |
|---|---|---|---|
| ZERO-SHOT | Edge Existence | **42.8** | 39.8 |
| | Node degree | 10.8 | **11.6** |
| | Node count | 5.4 | **6.6** |
| | Edge count | **5.6** | 5.4 |
| | Connected nodes | 1.6 | **3.4** |
| | Cycle Check | 65.2 | **84.4** |
| COT | Edge Existence | 50.4 | **50.8** |
| | Node degree | 8.8 | **10.0** |
| | Node count | **8.4** | 5.8 |
| | Edge count | 4.2 | **5.0** |
| | Connected nodes | **10.2** | 7.2 |
| | Cycle Check | **77.4** | 74.4 |

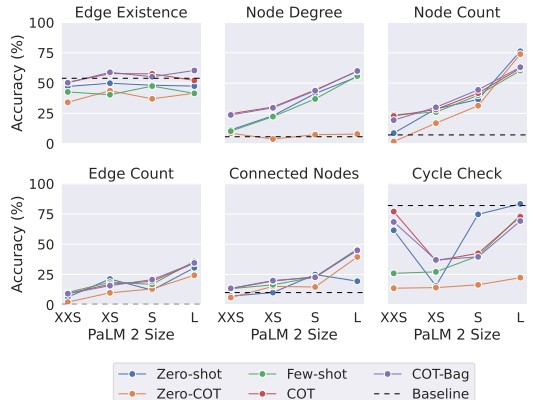

Table 3: Results on multiple relations for edge encoding with PaLM 2 XXS.

Figure 3: Effect of Model Capacity on graph reasoning task for PaLM 2-XXS, XS, S, and L.

for the concept of friendship and conducted experiments using two distinct question encoder functions: the graph question encoder and the application question encoder. The graph question encoder is responsible for encoding graph-related tasks, such as determining the degree of a specific node (*e.g.*, "What is the degree of node $i$?"). This encoder is used for obtaining results in Section 3.1. On the other hand, the application question encoder interprets graph questions in a more practical, day-to-day context. In the application scenario, we used a friendship-based scenario where we transformed the questions for each graph task as stated in Table 8.

**Results:** Table 2 summarizes the results of our experiment on question encoder functions. As the results show, the application encoder outperforms the graph encoder on almost all tasks, despite both encoders having the same graph encoder function and only differing slightly in how they ask the question. For example, on the ZERO-SHOT *edge existence* using PALM 2 XXS, the graph encoder obtained $42.8\%$ accuracy, while the application encoder obtained $60.8\%$.

**Summary:** The selection of the question encoder function affects the performance of LLMs when handling basic graph algorithms. As a result, it becomes important to translate a given task into more contextually meaningful textual information when employing LLMs for inference.

## 3.3 EXPERIMENT 3: MULTIPLE RELATION ENCODING

In this experiment setup, we introduce a modification to the friendship graph encoding function, which characterizes edges based on a range of distinct relation types, including *friends*, *colleagues*, *spouses*, *siblings*, *neighbors*, *acquaintances*, *teammates*, *classmates*, *coworkers*, or *roommates*. The selection of the relation type is randomized from this predefined set, thereby using multiple words to reference the existence of a relationship between nodes. This is a departure from using the same token(s) for edge representation in prior graph encoding experiments.

**Results:** As Table 3 shows, using multiple words to represent relationships did not hurt LLM performance and even improved it in some cases. This improvement is likely because the diverse set of relations provides the LLM with more textual information to perform the task, and the final encoding is closer to the text that the LLM may have seen during training, compared to the prior setup.

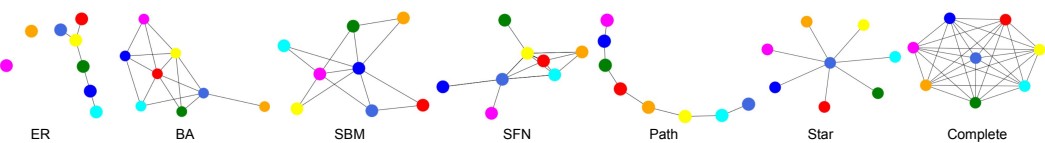

Figure 4: Samples of graphs generated with different graph generators from `GraphQA`.

## 3.4 Experiment 4: Model Capacity and Graph Reasoning Ability

In this experiment, we measure the effect of model capacity on the graph tasks. We compare the results of PaLM 2 (Anil et al., 2023) XXS, XS, S, and L, which have different number of parameters and therefore different capacity. We report the majority baseline for reference.

**Results: Model capacity has a significant effect on the graph reasoning ability of an LLM.** The results of this experiment, reported in Figure 3, show the larger model is generally better at graph reasoning tasks. This is because it has more capacity to learn and store complex information. The model capacity has less effect on *edge existence*. The results also show that the model was not able to beat the majority baseline for *edge existence* even with a large capacity.

## 3.5 Experiment 5: Reasoning in the Absence of Edges

In this experiment, we evaluate the performance of LLMs on *disconnected nodes*. In this task, we provide a graph description to the LLM, specifying the nodes and edges, and ask about the nodes that are *not* directly connected to a given node. This task differs from the previous ones in that it requires reasoning about information that is implicit in the graph, *i.e.*, information that is not explicitly mentioned in the output of the graph encoding function.

**Results: LLMs lack a global model of a graph.** The ZERO-SHOT prompting method achieved an accuracy of $0.5\%$, while the ZERO-COT, FEW-SHOT, COT, and COT-BAG methods achieved close to $0.0\%$ accuracy. These results suggest that LLMs perform significantly worse on the disconnected nodes task than on the connected nodes task. We believe that this is because the graph encoding functions primarily encode information about connected nodes, while not explicitly encoding information about nodes that are not connected. As a result, LLMs are better at processing relationships among connected nodes than at capturing the absence of connections, leading to sub-optimal performance in disconnectivity-related tasks.

## 4 Does the structure of the graph matter for the LLM?

It is natural to wonder if the structure of the graph itself might effect LLM's ability to reason over it. Inspired by recent work in analyzing graph neural networks (Palowitch et al., 2022; Yasir et al., 2023) this section seeks to measure a LLM's reasoning capabilities over graph with distinct structures. In this section, we show that graph structure can have significance influence on an LLM's reasoning performance. Figure 4 illustrates graphs created through different generative processes.

### 4.1 Random Graph Generation

To be able to experiment with LLMs on graphs, we generate random graphs using various graph generator algorithms. This allows us to: **Cover a wide range of properties.** Different graph generators produce graphs with different properties. For example, Erdős-Rényi graphs tend to be sparse and have a small average degree, while Barabási-Albert graphs tend to be dense and have a power-law degree distribution. By using a diverse set of generators, we ensure that the `GraphQA` benchmark includes graphs with a wide range of properties. **Avoid bias in graph problem evaluation.** The goal of generating such graphs is to test the ability of LLMs to solve graph problems. Graph problems can vary in difficulty depending on the properties of the graphs, so we use a diverse set of graphs to avoid bias. **Provide realistic benchmarks.** Real-world graphs exhibit a wide range of properties, and no single graph generator can capture all of these properties perfectly. By using a diverse set of generators, we create a benchmark that is more representative of real-world graphs. To generate random graphs, we use Erdős-Rényi (ER) graphs (Erdős & Rényi, 1959), scale-free net-

Table 4: Comparing graph generators with PaLM 62B. Underline and bold represent the most effective prompting heuristic and the top performing graph generator respectively.

| Method | Graph generator | Edge Existence | Node degree | Node count | Edge count | Connected nodes | Cycle check |
|---|---|---|---|---|---|---|---|
| ZERO-SHOT | Overall | 49.1 | 17.6 | 23.0 | 12.1 | 23.3 | 75.2 |
| | ER | 45.1 | 13.6 | 22.1 | 11.7 | 14.9 | 76.3 |
| | BA | 50.2 | 18.0 | 24.9 | 13.6 | 20.1 | 72.0 |
| | SBM | 45.0 | 13.8 | 21.9 | 9.2 | 13.8 | 86.5 |
| | Star | 58.0 | 34.0 | 32.8 | 31.7 | 61.7 | 8.1 |
| | SFN | 57.6 | 23.1 | 19.9 | 8.0 | 38.1 | 90.0 |
| | Path | **60.9** | 14.8 | 31.9 | 28.8 | 26.6 | 5.9 |
| | Complete | 19.8 | 12.6 | 20.7 | 6.2 | 13.3 | **91.7** |
| COT | Overall | 40.4 | 29.6 | 31.7 | 12.2 | 24.3 | 59.5 |
| | ER | 41.2 | 28.4 | 28.8 | 12.6 | 12.8 | 61.2 |
| | BA | 40.0 | 30.0 | 35.0 | 14.3 | 20.8 | 58.5 |
| | SBM | 40.3 | 26.5 | 30.2 | 8.7 | 13.0 | 65.8 |
| | Star | 40.3 | **38.0** | **41.8** | **31.6** | **68.6** | 21.3 |
| | SFN | 40.2 | 32.2 | 30.8 | 7.1 | 43.2 | 66.0 |
| | Path | 42.0 | 35.1 | 35.3 | 31.1 | 27.6 | 19.7 |
| | Complete | 39.6 | 21.9 | 28.9 | 3.9 | 14.6 | 69.3 |

works (SFN) (Barabási & Albert, 1999), Barabási–Albert (BA) model (Albert & Barabási, 2002), and stochastic block model (SBM) (Holland et al., 1983), in addition to star, path, and complete graph generators. We use NetworkX (Hagberg et al., 2008) to generate the random graphs. The details are reported in Appendix A.8.

## 4.2 RESULTS ON RANDOM GRAPH GENERATORS

Previous experiments have studied the performance of LLMs on basic graph tasks using random graphs generated using the Erdős-Rényi (ER) model. However, ER graphs often do not accurately represent the characteristics of real-world graphs. In this experiment, we investigate the effect of different random graph generators on the performance of LLMs on graph reasoning tasks. To make the experiment more realistic, we sample the few-shot examples randomly from graphs generated using different algorithms. We report the results of this experiment in Table 4.

**Graph structure has a significant impact on the LLM's performance.** The results show that the algorithm used to generate the graph has a significant impact on the performance of the LLM on graph tasks. For example, the cycle check task achieves 91.7% accuracy on complete graphs and 5.9% accuracy on path graphs. This is because the LLM has a strong prior towards graphs having cycles. Therefore, the accuracy is high for complete graphs, which always have cycles, and very low for path graphs, which never have cycles. By adding few-shot examples some having a cycle and some not, the accuracy of cycle check on path graphs increased from 5.9% to 19.7%. As another example, on *edge existence*, the LLM achieves 60.0% accuracy on path graphs, which are less likely to have an edge between two nodes, and 19.8% accuracy on complete graphs, which have edges between all pairs of nodes. This shows that the LLM has a prior that two nodes in a graph are more likely to be disconnected.

**Distractive statements in the graph encoding function disrupt the performance of the LLM.** The accuracy of node degree, node count, and connected nodes tasks is highest for star and path graphs. This is likely because the star and path graphs are more likely to have fewer edges and their graph encoding is most likely shorter with less distracting statements to these tasks. This is also evident from the accuracy of these tasks being among the lowest in complete graphs, which have many edges to specify and therefore many distractors.

**Adding out-of-distribution few-shot examples helped the LLM.** Similarly to the experiment in Section 3.1, adding few-shot examples and their chain of thought in COT prompting helped on most tasks. The key difference between the few-shot examples in this experiment and the previous one is that in this case, the examples are not required to come from the same graph generator algorithm. This shows that few-shot examples do not need to come from the same generator for the LLM to be helpful, and their main role is to explain the task to the LLM.

**Summary:** The performance of large language models (LLMs) on graph tasks is significantly impacted by the graph structure and the distracting statements in the graph encoding function. Graphs

with fewer edges and less complex encodings tend to perform better on most tasks. Adding few-shot examples, even if they are out-of-distribution, can help the LLM to perform better on most tasks.

# 5 RELATED WORK

**In-context learning.** One approach for reasoning with LLMs is to pre-train it on a large corpus of text that is closely related to the task. This has been shown to improve the performance (Hendrycks et al., 2021; Shen et al., 2021), but it can be computationally expensive, especially for larger models. Additionally, fine-tuning often demands domain-specific data and human expertise, adding to the cost. Brown et al. (2020b) has demonstrated the capabilities of LLMs in tackling novel tasks with little or no training data. The FEW-SHOT method inserts $k$ in-context input-output pairs before the test input and has been shown to significantly improve the performance of the LLM on unseen tasks. Recent research has proposed strategies to improve the selection of in-context demonstrations, such as retrieving semantically similar examples (Liu et al., 2021), employing chain-of-thought reasoning (Wei et al., 2022), and decomposing tasks into sub-problems using least-to-most prompting (Zhou et al., 2022a). In this work, we focus on evaluating and enhancing LLMs on basic graph reasoning tasks. We exploit some of the ideas in the literature and compare their results.

**Text-based reasoning with LLMs.** Numerous models have been proposed for text-based reasoning employing LLMs (see (Huang & Chang, 2022) for a survey). One approach to reasoning with LLMs is modular reasoning. This methodology divides the problem into smaller modules, utilizing distinct LMs to address each module (Zhou et al., 2022a; Kazemi et al., 2022; Khot et al., 2022). Another approach to reasoning with LLMs aims to predict the output of a question in a single LM call. This study primarily focuses on the latter method.

**Knowledge-Augmented LLMs.** Another body of work is concerned with the use of knowledge (frequently stored in *knowledge graphs* (KGs)) to improve LLM understanding of the world (Pan et al., 2023). Several different methodologies have been proposed which range from generating additional training data from KGs (Guu et al., 2020; Lewis et al., 2020; Agarwal et al., 2021) to extending pretraining (Yasunaga et al., 2022; Jin et al., 2023).

**Reasoning on graphs using LLMs.** The combination of graph learning and reasoning with LLMs is a rapidly growing area. InstructGLM (Ye et al., 2023) proposed an instruction-finetuned LLM for performing node classification. Chen et al. (2023) used LLMs as enhancers to exploit text attributes to be used in a graph learning model or as predictors for node classification on text-attributed graphs. Sanchez et al. (2023) encode text sequences into graphs (and back), and leverage LLMs as the mapping functions between them to aid commonsense reasoning. The closest work to ours is Wang et al. (2023), which proposed a set of tasks for benchmarking LLMs on graphs. However, this work omitted several natural graph tasks, lacked variety in the type of graph structure considered, and fixed the graph and question encoder function. They conclude that LLMs have preliminary graph reasoning abilities on somewhat complex graph tasks.

**Present work.** In this study, we focus on basic graph tasks, which are essential intermediate steps for more complex reasoning tasks on graphs. We conduct extensive experiments on graph and question encoder functions, as well as a wide range of graph generator functions. We provide an extensive study of graph encoding methods for black-box LLM usage, and introduce `GraphQA`, a new graph benchmark that illustrates the effect of graph structure on LLM encoding. We also provide insights and best practices for encoding graphs as text for use in LLMs.

# 6 CONCLUSIONS

In this work, we have presented the first comprehensive study of encoding graph-structured data as text for consumption by LLMs. We show that LLM performance on graph reasoning tasks varies on three fundamental levels: (1) the graph encoding method, (2) the nature of the graph task itself, and (3) interestingly, the very structure of the graph considered. These novel results provide valuable insight on strategies for encoding graphs as text – which can boost performance on graph reasoning tasks inside LLMs by 4.8% to 61.8%. We believe that this is a fruitful avenue for further investigation, and hope that our `GraphQA` benchmark tasks inspire additional work in the area.

## 7 ACKNOWLEDGEMENT

We express our sincere gratitude to Anton Tsitsulin, Dustin Zelle, Silvio Lattanzi, Vahab Mirrokni, and the entire graph mining team at Google Research, for their insightful comments, thorough proofreading, and constructive feedback which greatly enhanced the quality of our work. Furthermore, we extend our appreciation to the anonymous ICLR reviewers for their constructive suggestions. Their expertise and feedback played a crucial role in refining our paper.

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

## A   APPENDIX

### A.1   GRAPH ENCODING FUNCTION

We conducted an investigation into various methodologies for representing graphs as text. This process of encoding graphs as text can be separated into two key inquiries: First, the encoding of nodes within the graph, and second the encoding of edges between the nodes.

*Encoding Nodes*. Regarding the encoding of nodes, we examined several techniques, including:

- Integer encoding (*e.g.*, *Node 0*).

- Utilizing well-known English first names (*e.g.*, David).

- Utilizing popular character names in television series Game of Thrones and South Park.

- Incorporating the first names of American politicians.

- Employing alphabet letters for representation.

*Representing Edges*. Regarding the encoding of the edges, we examined the following techniques:

- Parenthesis: describing edges as (source node, target node).

- Friendship: source node and target node are friends.

- Coauthorship: source node and target node wrote a paper together.

- Social network: source node and target node are connected.

- Arrows: source node $\rightarrow$ target node.

- Incident: source node is connected to target nodes.

Combining the node and edge encoding, we start with the following list of graph encoding functions:

- **Adjacency.** using integer node encoding and parenthesis edge encoding.

- **Incident.** using integer node encoding and incident edge encoding.

- **Friendship**. using well-known english first names as node encoding and friendship edge encoding.

- **Co-authorship.** using well-known english first names as node encoding and coauthorship edge encoding.

- **SP.** using South Park character names as node encoding and friendship as edge encoding.

- **GOT.** using Game of Thrones character names as node encoding and friendship as edge encoding.

- **Social network.** using well-known English first names and social network edge encoding.

- **Politician.** using politician American politician first names and social network edge encoding.

- **Expert.** employing alphabet letters for node encoding and arrows as edge encoding. The encoding starts with "You are a graph analyst" (expert prompting (Zhang et al., 2023a)).

Here, we provide the full details for the graph encoding functions for the graph example in Figure 5.

**Adjacency:** In an undirected graph, (i,j) means that node i and node j are connected with an undirected edge. G describes a graph among nodes 0, 1, 2, 3, 4, 5, 6, 7, and 8.
The edges in G are: (0, 1) (0, 2) (1, 2) (2, 3) (2, 4) (2, 5) (2, 7) (3, 8) (5, 6) (6, 7) (7, 8).

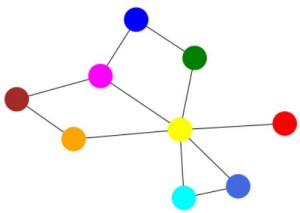

Figure 5: Running example graph for all graph encoding functions.

**Incident:** G describes a graph among 0, 1, 2, 3, 4, 5, 6, 7, and 8.
In this graph:
Node 0 is connected to nodes 1, 2.
Node 1 is connected to nodes 0, 2.
Node 2 is connected to nodes 0, 1, 3, 4, 5, 7.
Node 3 is connected to nodes 2, 8.
Node 4 is connected to node 2.
Node 5 is connected to nodes 2, 6.
Node 6 is connected to nodes 7, 5.
Node 7 is connected to nodes 2, 8, 6.
Node 8 is connected to nodes 3, 7.

**Co-authorship:** G describes a co-authorship graph among James, Robert, John, Michael, David, Mary, Patricia, Jennifer, and Linda.
In this co-authorship graph:
James and Robert wrote a paper together.
James and John wrote a paper together.
Robert and John wrote a paper together.
John and Michael wrote a paper together.
John and David wrote a paper together.
John and Mary wrote a paper together.
John and Jennifer wrote a paper together.
Michael and Linda wrote a paper together.
Mary and Patricia wrote a paper together.
Patricia and Jennifer wrote a paper together.
Jennifer and Linda wrote a paper together.

**Friendship:** G describes a friendship graph among James, Robert, John, Michael, David, Mary, Patricia, Jennifer, and Linda.
We have the following edges in G:
James and Robert are friends.
James and John are friends.
Robert and John are friends.
John and Michael are friends.
John and David are friends.
John and Mary are friends.
John and Jennifer are friends.
Michael and Linda are friends.
Mary and Patricia are friends.
Patricia and Jennifer are friends.
Jennifer and Linda are friends.

**SP:** G describes a friendship graph among Eric, Kenny, Kyle, Stan, Tolkien, Heidi, Bebe, Liane, and Sharon.
In this friendship graph:
Eric and Kenny are friends, Eric and Kyle are friends, Kenny and Kyle are friends, Kyle and Stan are friends, Kyle and Tolkien are friends, Kyle and Heidi are friends, Kyle and Liane are friends, Stan and Sharon are friends, Heidi and Bebe are friends, Bebe and Liane are friends, Liane and Sharon are friends.

**GOT:** G describes a friendship graph among Ned, Cat, Daenerys, Jon, Bran, Sansa, Arya, Cersei, and Jaime.
In this friendship graph: Ned and Cat are friends, Ned and Daenerys are friends, Cat and Daenerys are friends, Daenerys and Jon are friends, Daenerys and Bran are friends, Daenerys and Sansa are friends, Daenerys and Cersei are friends, Jon and Jaime are friends, Sansa and Arya are friends, Arya and Cersei are friends, Cersei and Jaime are friends.

**Social Network:** G describes a social network graph among James, Robert, John, Michael, David, Mary, Patricia, Jennifer, and Linda.
We have the following edges in G:
James and Robert are connected.
James and John are connected.
Robert and John are connected.
John and Michael are connected.
John and David are connected.
John and Mary are connected.
John and Jennifer are connected.
Michael and Linda are connected.
Mary and Patricia are connected.
Patricia and Jennifer are connected.
Jennifer and Linda are connected.

**Politician:** G describes a social network graph among Barack, Jimmy, Arnold, Bernie, Bill, Kamala, Hillary, Elizabeth, and John.
We have the following edges in G:
Barack and Jimmy are connected.
Barack and Arnold are connected.
Jimmy and Arnold are connected.
Arnold and Bernie are connected.
Arnold and Bill are connected.
Arnold and Kamala are connected.
Arnold and Elizabeth are connected.
Bernie and John are connected.
Kamala and Hillary are connected.
Hillary and Elizabeth are connected.
Elizabeth and John are connected.

**Expert:** You are a graph analyst and you have been given a graph G among A, B, C, D, E, F, G, H, and I. G has the following undirected edges:
A -> B
A -> C
B -> C
C -> D
C -> E
C -> F
C -> H
D -> I
F -> G
G -> H
H -> I

A.2 DETAILS ON GRAPHQA

A.2.1 GRAPH TASKS

GraphQA consists of a diverse set of basic graph problems, including:

- *Edge existence.* Determine whether a given edge exists in a graph.
- *Node degree.* Calculate the degree of a given node in a graph.
- *Node count.* Count the number of nodes in a graph.
- *Edge count.* Count the number of edges in a graph.
- *Connected nodes.* Find all the nodes that are connected to a given node in a graph.
- *Cycle check.* Determine whether a graph contains a cycle.
- *Disconnected nodes.* Find all the nodes that are not connected to a given node in a graph.
- *Reachability.* Determine whether there is a path from one node to another.
- *Shortest path.* Calculate the length of the shortest path from one node to another.
- *Maximum flow.* Calculate the maximum feasible flow from one node to another.
- *Triangle counting.* Count the number of triangles in the graph.
- *Node classification.* For this task. we used the stochastic block model graph generator to create two distinct blocks representing soccer and baseball enthusiasts. We then label a subset of nodes and ask the LLM to determine whether an unspecified node belonged to the soccer or baseball group. This exercise aimed to assess the LLM's ability to exploit the homophily in a given graph.

These tasks are all relatively simple, but they require LLMs to be able to reason about the relationships between nodes and edges in a graph. While adhering to basic graph tasks, we aimed for a diverse set of tasks, including discriminative (*e.g.*, *cycle check*) and generative (*e.g.*, *connected* or *disconnected nodes*) challenges. These tasks covered various aspects of graph analysis, from existence checks (*e.g.*, *edge existence*) to quantitative assessments (*e.g.*, *node count*), path analysis (*e.g.*, *cycle check*), recall-based tasks (*e.g.*, *connected nodes*), and null space exploration (*e.g.*, *disconnected nodes*).

The basic graph tasks listed above are all essential intermediate steps for more complex reasoning tasks on graphs. For example, to determine the shortest path between two nodes in a graph, we must first be able to find all the nodes that are connected to a given node. To detect communities in a graph, we must first be able to identify all the cycles in the graph. To find the most influential node in a graph, we must first be able to calculate the degree of each node. These tasks are essential building blocks for more complex reasoning tasks on graphs.

A.2.2 GRAPH GENERATION

To generate random graphs, we use Erdős-Rényi (ER) graphs (Erdős & Rényi, 1959), scale-free networks (SFN) (Barabási & Albert, 1999), Barabási–Albert (BA) model (Albert & Barabási, 2002), and stochastic block model (SBM) (Holland et al., 1983), in addition to star, path, and complete graph generators. Here, we briefly describe these graph generator algorithms:

- ER takes two parameters: the number of vertices $n$ and the probability of an edge existing between any two vertices $p$. For each pair of vertices, the generator randomly decides whether to create an edge between them with probability $p$. This process results in a graph with an average edge density of $p$.
- SFN graphs are a type of network whose degree distribution follows a power law. This means that a small number of nodes (hubs) have a very large number of connections, while the vast majority of nodes have very few connections. SFNs are found in many real-world networks, such as the World Wide Web, social networks, and biological networks.
- BA model creates scale-free networks by adding new nodes that connect to existing nodes with a probability proportional to their degree. In other words, new nodes are more likely to connect to nodes that already have many connections. This results in a network with a few highly connected nodes and many sparsely connected nodes.

- SBM is a generative model for random graphs that assumes the presence of underlying communities or groups of nodes within the network. It characterizes these communities by assigning a probability to each pair of nodes being connected, with higher probabilities for nodes within the same community and lower probabilities for nodes in different communities.

- A star graph is a type of graph in which one vertex, called the central vertex, is connected to all other vertices, which are called leaves. The central vertex has a degree of n-1, where n is the number of vertices in the graph. The leaves all have a degree of 1.

- A path graph is a type of graph in which the vertices can be listed in a linear order such that every edge connects two consecutive vertices in the list. In other words, a path graph is a simple chain of vertices connected by edges.

- A complete graph is a type of graph in which every pair of distinct vertices is connected by a unique edge.

To generate graphs, we sampled 500 graphs for each of the following algorithms: ER, BA, SFN, and SBM. We sampled 100 graphs for path, complete, and star graphs, as these have less variety. All graphs had between 5 and 20 nodes. For ER graphs, we sampled the probability for edge creation from $[0, 1]$. For SBM graphs, number of communities has been sampled from 2 to 10.

### A.2.3 GRAPH STATISTICS

Table 5 reports the statiscs of the graph generated. We are commited to open-source the code

Table 5: Ranking of graph encoding functions from experiment in Section 3.1 (lower better).

| Graph generator | Number of nodes | Number of edges | Node degree |
|---|---|---|---|
| ER | 12.37 | 39.79 | 5.70 |
| SFN | 12.31 | 23.28 | 3.70 |
| BA | 12.25 | 30.02 | 4.40 |
| SBM | 12.27 | 38.68 | 5.59 |
| Star | 11.60 | 10.60 | 1.80 |
| Path | 11.60 | 10.60 | 1.80 |
| Complete | 11.60 | 70.51 | 10.60 |

### A.3 AGGREGATED RESULTS ON GRAPH ENCODING FUNCTIONS

To provide recommendations about the best graph encoding function to use for each prompt type, we rank the encoders by their average standing (in rank order) on each graph task. The results are presented in Table 6, where a lower number is better (the encoder ranked higher on average). We note that for most prompting methods, incident encoding performed the best. However, for ZERO-SHOT graph prompting, node tokens with more established representations (such as politicians or popular fantasy characters) outperformed incident encoding. Table 7 reports the average (and standard deviation) of the reported results in Section 3.1 aggregated over the graph encoding function. In terms of the average results, incident encoding performed the best across all the tasks. We posit that the success of incident encoding can be attributed to two key factors. Firstly, it leverages integer node encoding (e.g., *node 0* or *node 1*), as we previously emphasized the advantages of this approach in Section 3.1.1. Secondly, incident edge encoding effectively captures the one-hop neighborhood around a graph, outperforming methods that simply list edges in a random order.

### A.4 EXPERIMENT 2: VARYING PROMPT QUESTION

In experiment 2 in the main body of the paper, we maintained the graph encoding function as a constant for the concept of friendship and conducted experiments using two distinct question encoder functions: the graph question encoder and the application question encoder. Table 8 states how the question encoder function varies for each graph task. In summary, *edge existence* became "assessing friendship existence", *node degree* became "counting the number of friends for an individual",

Table 6: Ranking of graph encoding functions from experiment in Section 3.1 (lower better).

| Encoding | ZERO-SHOT | ZERO-COT | FEW-SHOT | COT | COT-BAG |
|---|---|---|---|---|---|
| Adjacency | 4.83 | 3.25 | 2.16 | 3.00 | 1.83 |
| Incident | 6.16 | **2.58** | **2.00** | **2.33** | **1.33** |
| Co-authorship | 6.08 | 6.33 | 5.58 | 6.75 | 8.83 |
| Friendship | 5.16 | 6.41 | 6.25 | 4.66 | 6.00 |
| SP | 5.16 | 4.50 | 5.25 | 5.75 | 4.66 |
| GOT | 4.33 | 4.08 | 5.83 | 5.00 | 6.25 |
| Social Network | 4.58 | 6.50 | 5.83 | 6.16 | 6.41 |
| Politician | **3.50** | 6.33 | 6.25 | 5.58 | 4.00 |
| Expert | 5.16 | 5.00 | 5.83 | 5.75 | 5.66 |

Table 7: Aggregated results from experiment in Section 3.1 over graph encoding functions with mean and standard deviation (higher mean better).

| Graph Encoding | ZERO-SHOT | ZERO-COT | FEW-SHOT | COT | COT-BAG |
|---|---|---|---|---|---|
| Adjacency | 30.4 (21.5) | 20.8 (14.4) | 32.3 (13.8) | 45.9 (17.6) | 43.9 (16.4) |
| Incident | **35.6** (20.8) | **27.4** (12.4) | **36.6** (11.4) | **48.1** (18.7) | **49.6** (16.8) |
| Co-authorship | 28.3 (22.4) | 15.8 (10.1) | 21.1 (13.3) | 24.5 (17.9) | 18.4 (10.1) |
| Friendship | 29.5 (27.2) | 16.1 (11.0) | 21.5 (14.3) | 27.2 (20.1) | 24.4 (16.1) |
| SP | 29.9 (26.1) | 17.6 (11.8) | 21.3 (11.4) | 25.9 (18.7) | 24.5 (14.5) |
| GOT | 30.9 (25.4) | 17.7 (12.0) | 19.3 (12.1) | 26.2 (19.3) | 23.6 (15.7) |
| Social Network | 30.4 (25.5) | 15.1 (10.0) | 21.0 (11.3) | 26.3 (19.2) | 24.7 (16.9) |
| Politician | 30.9 (25.3) | 15.3 (12.2) | 20.2 (11.1) | 26.6 (18.4) | 26.1 (17.5) |
| Expert | 29.3 (20.6) | 17.6 (9.4) | 18.8 (6.4) | 24.7 (13.2) | 23.8 (10.9) |

*node count* became "counting the number of people mentioned", *edge count* became "counting the number of friendships mentioned", and *connected nodes* became "listing friends".

Table 8: Examples for graph tasks and their corresponding graph question encoder and application question encoder.

| Task | Graph question encoder | Application question encoder |
|---|---|---|
| *Edge existence* | Is node Christopher connected to node Michael? | Are Christopher and Michael friends? |
| *Node degree* | What is the degree of node 14? | How many friends does Christopher have? |
| *Node count* | How many nodes are in this graph? | How many people are mentioned in this information? |
| *Edge count* | How many edges are in this graph? | How many friendships are given in this information? |
| *Connected nodes* | List all the nodes connected to node Christopher. | List all the people who are friends with Christopher. |

## A.5 MORE COMPLEX GRAPH REASONING TASKS

In this experiment, we expand our experiments with GraphQA to encompass a wider range of complex graph reasoning tasks, including *reachability*, *shortest path*, *maximum flow*, *triangle counting*, and *node classification*. Results from these additional tasks are shown in Table 9. Our core findings remain consistent: the chosen graph encoding function plays a crucial role in influencing the LLM's graph reasoning performance. For instance, in *triangle counting*, the social network encoding yields $7.0\%$ accuracy, while the adjacency encoding resulted in a significantly higher $27.0\%$ accuracy. The main results hold for these complex tasks as well. Mainly, the graph encoding function has a significant impact on LLM graph reasoning *e.g.*, *triangle counting* gets $7.0\%$ with social network and $27.0\%$ with adjacency.

## A.6 EXPERIMENTING WITH GPT-3.5-TURBO

In this section, we conduct experiments using gpt-3.5-turbo (Brown et al., 2020a) and report the results in Table 10. This additional evaluation provides further insights into the performance of

Table 9: Comparison of various graph encoding functions based on their accuracy on different graph tasks using PaLM 62B. The most effective prompting heuristic is highlighted with an underline, and the top-performing graph encoding function for it is highlighted in bold.

| Method | Encoding | Reachability | Shortest path | Maximum flow | Triangle counting | Node classification |
|---|---|---|---|---|---|---|
| ZERO-SHOT | Overall | 84.9 | 11.5 | 12.2 | 1.5 | 12.0 |
| | Adjacency | 83.8 | 10.2 | 12.0 | 2.0 | 16.4 |
| | Incident | 84.6 | 9.8 | 12.2 | 1.2 | 15.8 |
| | Co-authorship | 84.2 | 18.0 | 12.8 | 1.8 | 9.2 |
| | Friendship | 83.8 | 11.0 | 12.2 | 1.6 | 3.0 |
| | SP | 85.4 | 12.4 | 13.2 | 0.8 | 10.8 |
| | GOT | 86.8 | 9.6 | 11.4 | 0.8 | 4.6 |
| | Social network | 85.8 | 13.0 | 12.8 | 2.0 | 5.4 |
| | Politician | 81.6 | 13.2 | 11.8 | 1.2 | 24.4 |
| | Expert | **87.8** | 6.4 | 11.4 | 1.8 | 18.4 |
| ZERO-COT | Overall | 73.5 | 33.6 | 37.9 | 12.7 | 77.4 |
| | Adjacency | 72.4 | 37.2 | 50.2 | **27.0** | 79.8 |
| | Incident | 77.2 | 35.4 | 56.0 | 25.6 | 80.6 |
| | Co-authorship | 71.6 | 35.8 | 33.4 | 9.2 | 73.6 |
| | Friendship | 75.0 | 32.4 | 32.4 | 8.6 | 81.0 |
| | SP | 79.8 | 39.6 | 33.2 | 10.2 | 68.6 |
| | GOT | 77.0 | 34.8 | 33.4 | 8.2 | 70.8 |
| | Social network | 79.8 | 29.4 | 36.4 | 7.0 | **83.4** |
| | Politician | 52.6 | 26.8 | 34.6 | 8.2 | 76.4 |
| | Expert | 76.4 | 31.4 | 31.8 | 10.4 | 82.2 |
| FEW-SHOT | Overall | 79.4 | 22.7 | 10.8 | 3.0 | 43.4 |
| | Adjacency | 76.8 | 19.8 | 12.2 | 3.8 | 34.2 |
| | Incident | 65.0 | 20.8 | 13.2 | 2.4 | 43.2 |
| | Co-authorship | 80.8 | 24.0 | 12.0 | 3.4 | 46.2 |
| | Friendship | 82.6 | 29.6 | 10.4 | 3.4 | 50.6 |
| | SP | 82.4 | 22.2 | 11.8 | 2.6 | 49.2 |
| | GOT | 82.0 | 20.6 | 10.2 | 2.8 | 49.4 |
| | Social network | 82.2 | 26.2 | 10.4 | 2.2 | 47.8 |
| | Politician | 82.4 | 24.6 | 9.2 | 3.2 | 49.6 |
| | Expert | 80.8 | 16.8 | 8.2 | 3.4 | 20.8 |
| COT | Overall | 45.2 | 38.6 | 45.0 | 8.1 | 54.4 |
| | Adjacency | 53.8 | 36.4 | 49.2 | 21.6 | 53.6 |
| | Incident | 39.4 | 35.2 | 47.2 | 19.8 | 54.4 |
| | Co-authorship | 52.0 | 43.8 | 43.8 | 5.6 | 50.2 |
| | Friendship | 40.2 | 46.6 | 41.4 | 5.0 | 57.2 |
| | SP | 46.2 | 37.4 | 43.2 | 4.2 | 60.2 |
| | GOT | 48.2 | 35.4 | 41.6 | 4.2 | 56.4 |
| | Social network | 43.8 | 42.8 | 44.8 | 3.2 | 56.0 |
| | Politician | 45.8 | 39.2 | 42.2 | 4.0 | 55.8 |
| | Expert | 37.6 | 30.2 | **51.8** | 5.6 | 46.0 |
| COT-BAG | Overall | 45.2 | 40.4 | 44.4 | 8.1 | 52.1 |
| | Adjacency | 53.8 | 38.0 | 49.6 | 20.8 | 50.2 |
| | Incident | 39.4 | 38.2 | 44.6 | 19.4 | 52.6 |
| | Co-authorship | 52.0 | 45.6 | 39.6 | 4.6 | 52.6 |
| | Friendship | 40.2 | **48.4** | 42.4 | 4.6 | 53.0 |
| | SP | 46.2 | 35.8 | 42.0 | 4.8 | 55.4 |
| | GOT | 48.2 | 40.0 | 42.0 | 4.0 | 55.2 |
| | Social network | 43.8 | 44.0 | 46.8 | 3.2 | 51.8 |
| | Politician | 45.8 | 40.6 | 41.4 | 4.0 | 54.4 |
| | Expert | 37.6 | 33.4 | 51.6 | 7.2 | 43.8 |

our approach and confirms the key findings observed on Palm 1 and 2. In particular, graph encoding functions have a substantial impact on the performance of LLMs on graph-based tasks, with the node degree task exhibiting a performance variance of 5.4% to 66.4%. Integer node encoding enhances arithmetic performance for node degree, node count, and edge count tasks. The incident encoding outperforms other encoding functions by a significant margin on the connected node task due to its ability to make information more readily accessible for LLM utilization.

Table 10: Comparing different graph encoding functions on different graph tasks for gpt-3.5-turbo on ZERO-SHOT prompting. The top-performing graph function encoder for is highlighted in bold.

| Encoding function | Edge Existence | Node degree | Node count | Edge count | Connected nodes | Cycle check |
|---|---|---|---|---|---|---|
| Overall | 77.0 | 42.2 | 98.8 | 37.6 | 46.7 | 86.1 |
| Adjacency | 73.0 | 5.4 | 99.2 | **46.2** | 59.2 | 84.2 |
| Incident | 80.2 | **66.4** | 96.2 | 12.6 | **75.2** | 84.6 |
| Co-authorship | **82.0** | 42.6 | 99.8 | 39.4 | 37.2 | 86.4 |
| Friendship | 73.8 | 43.8 | **100.0** | 41.8 | 41.2 | 87.4 |
| SP | 74.0 | 44.2 | 99.8 | 39.0 | 38.6 | 86.4 |
| GOT | 75.8 | 41.0 | 99.0 | 39.4 | 38.6 | **88.6** |
| Social Network | 78.6 | 47.6 | 99.8 | 38.6 | 40.6 | 86.6 |
| Politician | 78.6 | 45.4 | **100.0** | 39.8 | 40.2 | 84.2 |
| Expert | 75.7 | 43.5 | 95.3 | 41.7 | 49.2 | 86.6 |

## A.7 COMPARISON WITH EXISTING WORKS

In this section, we highlight the contributions of this paper by comparing it to some of the existing works in the literature. Our work uniquely investigates how properties of the graph structure (via synthetic generation) influence the choice of graph encoding function. This is completely novel, as no other related work examines these interactions. This novel approach enables us to systematically study these interactions, which have not been explored in previous research. Additionally, we conducted extensive scaling experiments with LLMs of varying parameter sizes which no other work has done. Finally, we have significant differences from each work individually, for example: unlike (Guo et al., 2023), we study a diversity of LLMs (and have added more models in the rebuttal), unlike (Chen et al., 2023) we vary prompting and graph tasks, and unlike (Ye et al., 2023), we use a black box model without model weights. Furthermore, we conducted experiments on varying question encoder functions to evaluate their impact on performance. Table 11 clearly states the novelty of our work.

Table 11: Comparing this work to existing works in the literature.

|  | Guo et al. (2023) | Chen et al. (2023) | Ye et al. (2023) | Ours |
|---|---|---|---|---|
| Synthetic generation | ✗ | ✗ | ✗ | ✓ |
| Black-box model | ✓ | ✓ | ✗ | ✓ |
| Scaling experiments | ✗ | ✗ | ✗ | ✓ |
| Varying question encoding function | ✗ | ✗ | ✗ | ✓ |
| Varying graph encoding function | ✓ | ✗ | ✗ | ✓ |
| Varying node encoding function | ✗ | ✗ | ✗ | ✓ |
| Varying edge encoding function | ✗ | ✗ | ✗ | ✓ |
| Varying graph structure | ✗ | ✗ | ✗ | ✓ |

## A.8 IMPLEMENTATION DETAILS

For our experiments, we used PaLM 62B and PaLM 2 (various sizes) served on a $4 \times 4$ TPU v4 architecture. The decoding temperature was set to zero. GPT results are obtained by connecting to the OpenAI API. We used the NetworkX library (Hagberg et al., 2008) to generate the random graphs and to find the answers to the graph tasks. We are committed to open-sourcing both our code and data upon the acceptance of our paper.

## A.9 EVALUATING MORE LLMS FOR GRAPH TASKS WITH DIFFERENT GRAPH ENCODING FUNCTIONS

We compared different graph encoding functions on a PaLM 62B (Chowdhery et al., 2022) in Section 3.1. Here, we provide the results of the same experiment on PaLM 2 XXS, XS, S, and L (Anil et al., 2023) in Tables 12, 14 and 16. We also provide results for some instruction-finetuned Flan (Chung et al., 2022) checkpoints of the same models in Tables 13 and 15.

Table 12: Comparing different graph encoding functions on different graph tasks for PaLM 2 XXS. The most effective prompting heuristic is highlighted with an underline, and the top-performing graph function encoder for the respective heuristic is highlighted in bold.

| Method | Encoding function | Edge Existence | Node degree | Node count | Edge count | Connected nodes | Cycle check |
|---|---|---|---|---|---|---|---|
| ZERO-SHOT | Overall | 47.2 | 11.3 | 8.7 | 6.4 | 7.2 | 61.5 |
| | Adjacency | 48.4 | 14.4 | 6.2 | 4.0 | 17.6 | 82.6 |
| | Incident | 45.2 | 13.4 | 7.2 | 5.2 | 11.2 | 68.4 |
| | Co-authorship | 45.4 | 10.8 | 7.4 | 4.6 | 5.2 | 66.4 |
| | Friendship | 42.8 | 10.8 | 5.4 | 5.6 | 1.6 | 65.2 |
| | SP | 56.6 | 11.0 | 7.2 | 5.8 | 3.0 | 26.6 |
| | GOT | 56.4 | 7.8 | 6.0 | 7.0 | 2.0 | 51.8 |
| | Social Network | 51.2 | 11.0 | 7.8 | 5.4 | 5.2 | 74.4 |
| | Politician | 40.6 | 12.0 | 9.4 | 6.8 | 10.0 | 73.2 |
| | Expert | 38.0 | 10.4 | 21.4 | 12.8 | 9.0 | 45.2 |
| ZERO-COT | Overall | 34.1 | 8.6 | 1.7 | 2.2 | 6.0 | 13.7 |
| | Adjacency | 20.2 | 19.0 | 2.4 | 2.0 | 9.0 | 16.4 |
| | Incident | 45.0 | 36.0 | 1.2 | 6.0 | 16.0 | 37.8 |
| | Co-authorship | 48.8 | 22.0 | 0.8 | 4.4 | 11.8 | 31.6 |
| | Friendship | 43.2 | 0.2 | 0.6 | 1.6 | 2.0 | 10.8 |
| | SP | 30.8 | 0 | 1.2 | 0.6 | 1.0 | 3.0 |
| | GOT | 21.8 | 0 | 1.2 | 0.6 | 2.6 | 4.8 |
| | Social Network | 39.4 | 0 | 5.0 | 1.8 | 4.8 | 6.2 |
| | Politician | 40.6 | 0 | 2.2 | 2.6 | 5.2 | 7.0 |
| | Expert | 16.8 | 0 | 0.4 | 0.6 | 1.6 | 6.0 |
| FEW-SHOT | Overall | 42.7 | 10.3 | 23.9 | 10.2 | 13.3 | 26.0 |
| | Adjacency | 50.2 | 11.8 | **77.6** | **27.0** | 17.4 | 83.4 |
| | Incident | 46.6 | 12.8 | 58.4 | 19.8 | 18.4 | 57.8 |
| | Co-authorship | 44.2 | 7.6 | 31.0 | 11.8 | 11.4 | 31.0 |
| | Friendship | 42.8 | 9.6 | 8.8 | 7.4 | 11.8 | 7.2 |
| | SP | 29.4 | 10.4 | 9.6 | 4.6 | 11.6 | 7.0 |
| | GOT | 26.0 | 10.0 | 8.2 | 5.4 | 9.0 | 9.8 |
| | Social Network | 40.4 | 9.4 | 8.4 | 4.2 | 12.0 | 11.2 |
| | Politician | 50.6 | 8.2 | 7.2 | 6.0 | 12.6 | 12.0 |
| | Expert | 54.0 | 12.6 | 6.0 | 6.0 | 15.8 | 14.6 |
| COT | Overall | 50.6 | 24.7 | 22.8 | 9.3 | 13.3 | 77.0 |
| | Adjacency | 51.0 | **80.8** | 72.2 | 22.0 | 19.6 | **84.0** |
| | Incident | 48.6 | 55.0 | 54.4 | 17.2 | 17.2 | 81.6 |
| | Co-authorship | 51.4 | 31.2 | 29.8 | 10.0 | 12.2 | 80.6 |
| | Friendship | 50.4 | 8.8 | 8.4 | 4.2 | 10.2 | 77.4 |
| | SP | 52.2 | 9.4 | 10.0 | 6.6 | 12.0 | 74.6 |
| | GOT | 51.4 | 9.2 | 8.0 | 5.6 | 10.4 | 70.4 |
| | Social Network | **53.8** | 8.6 | 7.8 | 5.8 | 8.4 | 76.0 |
| | Politician | 47.0 | 9.4 | 8.4 | 6.4 | 13.8 | 75.8 |
| | Expert | 49.6 | 10.2 | 6.4 | 5.6 | 15.6 | 72.6 |
| COT-BAG | Overall | 50.3 | 23.7 | 19.4 | 9.2 | 13.6 | 68.4 |
| | Adjacency | 49.6 | 73.0 | 57.8 | 22.8 | **17.6** | 82.4 |
| | Incident | 49.6 | 53.4 | 46.6 | 17.6 | 14.4 | 77.2 |
| | Co-authorship | 50.4 | 30.4 | 28.4 | 10.2 | 14.6 | 74.0 |
| | Friendship | 48.8 | 8.4 | 6.2 | 5.2 | 9.2 | 65.8 |
| | SP | 50.4 | 7.0 | 6.8 | 5.0 | 12.4 | 61.2 |
| | GOT | 51.8 | 11.0 | 6.0 | 5.0 | 13.2 | 57.2 |
| | Social Network | 55.8 | 10.6 | 9.4 | 4.6 | 11.0 | 59.4 |
| | Politician | 49.2 | 9.4 | 6.0 | 6.6 | 16.0 | 69.6 |
| | Expert | 47.4 | 10.2 | 7.4 | 6.2 | 14.2 | 68.6 |

Table 13: Comparing different graph encoding functions on different graph tasks for Flan-PaLM 2 XXS. The most effective prompting heuristic is highlighted with an underline, and the top-performing graph function encoder for the respective heuristic is highlighted in bold.

| Method | Encoding function | Edge Existence | Node degree | Node count | Edge count | Connected nodes | Cycle check |
|---|---|---|---|---|---|---|---|
| ZERO-SHOT | Overall | 56.6 | 11.1 | 11.7 | 3.9 | 8.2 | 20.8 |
| | Adjacency | 57.6 | 10.4 | 10.0 | 4.6 | 15.6 | 23.4 |
| | Incident | 59.8 | 12.2 | 11.8 | 4.2 | 11.6 | 32.4 |
| | Co-authorship | 58.2 | 10.8 | 9.8 | 5.0 | 6.8 | 25.8 |
| | Friendship | 57.2 | 11.0 | 11.6 | 2.6 | 3.0 | 18.4 |
| | SP | 54.8 | 11.4 | 15.4 | 3.0 | 5.8 | 17.0 |
| | GOT | 51.2 | 9.0 | 16.2 | 3.4 | 5.2 | 16.6 |
| | Social Network | 54.4 | 11.8 | 11.0 | 3.2 | 6.6 | 15.2 |
| | Politician | 55.8 | 12.2 | 8.0 | 4.8 | 7.6 | 18.4 |
| | Expert | **60.2** | 11.0 | 11.8 | 4.0 | 11.4 | 19.6 |
| ZERO-COT | Overall | 45.9 | 17.6 | 11.2 | 9.7 | 14.4 | 33.9 |
| | Adjacency | 51.2 | 26.2 | 5.2 | **16.6** | 19.2 | 70.6 |
| | Incident | 52.4 | 38.8 | 6.0 | 13.6 | 16.8 | 46.4 |
| | Co-authorship | 47.0 | 25.0 | 10.8 | 12.4 | 13.8 | 32.8 |
| | Friendship | 47.2 | 10.0 | 16.6 | 9.2 | 11.4 | 21.4 |
| | SP | 39.4 | 9.0 | 12.8 | 9.2 | 14.2 | 22.6 |
| | GOT | 40.8 | 9.4 | 11.6 | 7.8 | 11.0 | 17.4 |
| | Social Network | 47.2 | 10.0 | 12.4 | 7.2 | 11.6 | 15.2 |
| | Politician | 48.2 | 13.0 | 13.6 | 6.2 | 12.8 | 22.8 |
| | Expert | 39.8 | 16.8 | 11.6 | 4.8 | 19.0 | 56.2 |
| FEW-SHOT | Overall | 54.1 | 10.0 | 11.9 | 4.9 | 8.6 | 82.6 |
| | Adjacency | 53.2 | 11.2 | 16.2 | 4.6 | 9.2 | 82.8 |
| | Incident | 53.2 | 11.4 | 23.0 | 5.6 | 9.0 | 80.2 |
| | Co-authorship | 54.2 | 10.8 | 13.0 | 4.6 | 8.4 | 84.4 |
| | Friendship | 56.0 | 8.4 | 9.4 | 4.8 | 8.4 | 81.0 |
| | SP | 59.0 | 9.0 | 10.4 | 5.4 | 8.8 | 84.2 |
| | GOT | 52.8 | 8.8 | 9.4 | 5.8 | 8.6 | 84.6 |
| | Social Network | 50.6 | 9.8 | 8.8 | 4.2 | 7.8 | 85.4 |
| | Politician | 51.4 | 7.6 | 7.8 | 5.2 | 9.4 | 80.2 |
| | Expert | 56.6 | 12.8 | 9.2 | 3.8 | 8.0 | 80.8 |
| COT | Overall | 56.4 | 20.0 | 17.3 | 6.6 | 8.0 | 82.7 |
| | Adjacency | 57.4 | **51.6** | 31.8 | 14.8 | 10.4 | 80.0 |
| | Incident | 56.4 | 41.0 | 37.0 | 10.8 | 11.2 | 80.8 |
| | Co-authorship | 54.4 | 28.6 | 19.6 | 7.2 | 7.6 | 83.4 |
| | Friendship | 58.0 | 11.0 | 11.6 | 4.0 | 6.0 | 81.6 |
| | SP | 62.4 | 10.4 | 12.2 | 3.4 | 5.4 | 84.4 |
| | GOT | 54.4 | 11.2 | 13.2 | 4.2 | 5.4 | 84.0 |
| | Social Network | 56.0 | 8.2 | 12.4 | 3.8 | 6.0 | **86.0** |
| | Politician | 53.8 | 8.4 | 9.6 | 6.4 | 8.2 | 80.8 |
| | Expert | 55.2 | 9.8 | 8.6 | 5.0 | 11.6 | 83.0 |
| COT-BAG | Overall | 52.6 | 19.6 | 17.7 | 8.1 | 8.1 | 82.1 |
| | Adjacency | 53.4 | 50.2 | 30.4 | 17.4 | 10.2 | 81.2 |
| | Incident | 49.4 | 44.0 | **33.2** | 16.6 | 9.0 | 79.8 |
| | Co-authorship | 52.4 | 26.4 | 22.6 | 8.4 | 7.0 | 85.2 |
| | Friendship | 50.8 | 10.2 | 12.6 | 4.6 | 5.2 | 80.0 |
| | SP | 56.6 | 8.8 | 13.4 | 4.0 | 5.8 | 82.4 |
| | GOT | 52.6 | 11.0 | 12.8 | 5.4 | 6.4 | 82.4 |
| | Social Network | 54.0 | 8.0 | 12.6 | 5.4 | 7.8 | 82.2 |
| | Politician | 51.0 | 9.2 | 10.8 | 5.0 | 9.2 | 83.0 |
| | Expert | 53.2 | 9.0 | 10.6 | 5.8 | 12.6 | 82.8 |

Table 14: Comparing different graph encoding functions on different graph tasks for PaLM 2 XS. The most effective prompting heuristic is highlighted with an underline, and the top-performing graph function encoder for the respective heuristic is highlighted in bold.

| Method | Encoding function | Edge Existence | Node degree | Node count | Edge count | Connected nodes | Cycle check |
|---|---|---|---|---|---|---|---|
| ZERO-SHOT | Overall | 49.9 | 23.0 | 28.7 | 21.3 | 10.1 | 15.6 |
| | Adjacency | 50.8 | 22.4 | 11.4 | 22.2 | 25.8 | 21.8 |
| | Incident | 50.6 | 36.6 | 11.2 | 11.0 | 31.0 | 30.6 |
| | Co-authorship | 48.2 | 21.4 | 31.0 | 17.8 | 11.6 | 6.8 |
| | Friendship | 49.2 | 20.6 | 36.2 | 24.0 | 4.2 | 0 |
| | SP | 52.4 | 22.6 | 38.4 | **25.8** | 2.8 | 0.4 |
| | GOT | 53.8 | 17.8 | 32.2 | **25.8** | 1.8 | 0 |
| | Social Network | 44.6 | 22.4 | 35.8 | 24.4 | 2.0 | 0 |
| | Politician | 49.0 | 21.4 | 32.8 | 21.8 | 5.2 | 15.2 |
| | Expert | 50.2 | 22.0 | 29.6 | 18.6 | 6.6 | 65.4 |
| ZERO-COT | Overall | 43.7 | 3.8 | 16.9 | 9.9 | 15.2 | 14.2 |
| | Adjacency | 48.2 | 16.6 | 3.0 | 18.4 | 26.6 | 40.4 |
| | Incident | 53.6 | 10.6 | 2.6 | 6.2 | 50.8 | 43.6 |
| | Co-authorship | 46.8 | 2.6 | 9.2 | 2.2 | 19.6 | 8.6 |
| | Friendship | 32.8 | 0.4 | 18.4 | 12.6 | 3.2 | 3.6 |
| | SP | 40.2 | 0.4 | 21.4 | 4.0 | 4.8 | 7.6 |
| | GOT | 41.2 | 0.2 | 20.8 | 3.8 | 3.6 | 2.8 |
| | Social Network | 47.2 | 0 | 25.4 | 12.4 | 3.8 | 1.4 |
| | Politician | 47.0 | 0.8 | 27.6 | 15.6 | 4.8 | 10.0 |
| | Expert | 36.0 | 3.0 | 23.8 | 14.0 | 19.4 | 10.2 |
| FEW-SHOT | Overall | 40.4 | 22.3 | 26.0 | 18.7 | 16.5 | 27.2 |
| | Adjacency | 42.2 | 23.2 | 43.2 | 29.6 | 21.4 | 58.0 |
| | Incident | 48.6 | 35.4 | 58.8 | 31.8 | 34.0 | 41.2 |
| | Co-authorship | 42.6 | 24.0 | 22.2 | 18.2 | 15.2 | 29.4 |
| | Friendship | 45.0 | 17.4 | 16.8 | 13.8 | 13.2 | 18.2 |
| | SP | 36.6 | 23.6 | 16.2 | 16.0 | 13.0 | 20.2 |
| | GOT | 32.4 | 19.6 | 17.2 | 17.8 | 10.6 | 17.0 |
| | Social Network | 41.6 | 20.8 | 18.4 | 14.4 | 12.8 | 21.0 |
| | Politician | 43.0 | 16.2 | 17.6 | 12.8 | 11.6 | 26.0 |
| | Expert | 31.4 | 20.2 | 23.2 | 13.8 | 17.0 | 13.4 |
| COT | Overall | 57.8 | 30.2 | 28.2 | 17.0 | 19.7 | 36.4 |
| | Adjacency | 43.4 | 63.0 | 43.0 | 26.8 | 33.0 | 69.8 |
| | Incident | 55.8 | **63.8** | 54.4 | 24.6 | 44.2 | 38.8 |
| | Co-authorship | 59.6 | 27.2 | 25.2 | 13.4 | 17.2 | 40.6 |
| | Friendship | 64.2 | 19.0 | 20.2 | 12.8 | 13.0 | 40.8 |
| | SP | 62.0 | 19.2 | 18.0 | 16.6 | 15.0 | 10.2 |
| | GOT | 62.4 | 19.6 | 20.6 | 17.4 | 12.2 | 6.2 |
| | Social Network | 61.0 | 21.4 | 23.0 | 13.2 | 10.4 | 42.6 |
| | Politician | 55.2 | 18.4 | 21.4 | 14.0 | 13.6 | 61.4 |
| | Expert | 56.4 | 20.0 | 27.6 | 14.4 | 18.8 | 17.4 |
| COT-BAG | Overall | 58.9 | 29.6 | 30.0 | 15.8 | 20.0 | 37.1 |
| | Adjacency | 49.8 | 57.8 | 43.0 | 26.4 | 32.8 | **71.2** |
| | Incident | 57.4 | 61.8 | **50.0** | 23.8 | **41.8** | 55.8 |
| | Co-authorship | 59.0 | 27.0 | 25.8 | 15.6 | 17.6 | 34.8 |
| | Friendship | **66.2** | 22.6 | 22.6 | 10.0 | 10.2 | 38.2 |
| | SP | 61.2 | 18.4 | 23.8 | 15.2 | 13.4 | 15.6 |
| | GOT | 61.2 | 20.4 | 27.2 | 15.0 | 13.6 | 9.8 |
| | Social Network | 60.6 | 19.2 | 24.8 | 10.8 | 13.4 | 35.8 |
| | Politician | 54.8 | 17.8 | 23.2 | 11.2 | 15.4 | 53.6 |
| | Expert | 60.0 | 21.4 | 29.6 | 14.4 | 21.8 | 19.2 |

Table 15: Comparing different graph encoding functions on different graph tasks for Flan-PaLM 2 XS. The most effective prompting heuristic is highlighted with an underline, and the top-performing graph function encoder for the respective heuristic is highlighted in bold.

| Method | Encoding function | Edge Existence | Node degree | Node count | Edge count | Connected nodes | Cycle check |
|--------|-------------------|----------------|-------------|------------|------------|-----------------|-------------|
| ZERO-SHOT | Overall | 68.4 | 10.2 | 26.8 | 4.4 | 23.0 | 84.4 |
| | Adjacency | 78.0 | 17.6 | 39.0 | 7.2 | 34.4 | 87.2 |
| | Incident | 76.2 | 29.6 | 46.0 | 3.8 | 45.8 | 84.4 |
| | Co-authorship | 64.8 | 11.2 | 23.6 | 3.4 | 20.8 | 84.6 |
| | Friendship | 63.4 | 5.4 | 23.4 | 3.2 | 15.2 | 84.0 |
| | SP | 59.2 | 5.4 | 16.8 | 2.8 | 16.0 | 84.0 |
| | GOT | 62.6 | 4.6 | 19.6 | 3.2 | 18.2 | 83.2 |
| | Social Network | 72.0 | 4.4 | 17.6 | 4.0 | 17.8 | 84.0 |
| | Politician | 69.0 | 5.6 | 20.0 | 4.4 | 17.2 | 84.6 |
| | Expert | 70.6 | 7.6 | 35.4 | 7.2 | 21.4 | 84.0 |
| ZERO-COT | Overall | 54.3 | 16.4 | 32.2 | 13.4 | 25.1 | 59.9 |
| | Adjacency | 68.6 | 34.8 | 23.0 | 16.6 | 36.8 | 82.0 |
| | Incident | 59.4 | 51.2 | 24.2 | 11.0 | **55.8** | 67.4 |
| | Co-authorship | 51.8 | 15.0 | 25.8 | 11.8 | 26.6 | 41.4 |
| | Friendship | 53.8 | 6.2 | 41.6 | 12.2 | 19.6 | 70.6 |
| | SP | 49.4 | 5.8 | 33.6 | 13.8 | 14.4 | 37.0 |
| | GOT | 47.0 | 7.6 | 27.6 | 12.8 | 16.6 | 38.6 |
| | Social Network | 51.4 | 8.0 | 34.4 | 12.4 | 18.4 | 74.0 |
| | Politician | 55.6 | 8.8 | 35.6 | 12.8 | 14.6 | 53.8 |
| | Expert | 51.8 | 10.6 | **44.2** | 17.0 | 23.2 | 74.6 |
| FEW-SHOT | Overall | 70.9 | 13.2 | 21.4 | 10.0 | 10.4 | 87.2 |
| | Adjacency | 72.0 | 22.4 | 33.2 | 14.4 | 12.2 | 86.6 |
| | Incident | 81.8 | 27.0 | 33.6 | 7.2 | 22.0 | 83.4 |
| | Co-authorship | 68.0 | 17.8 | 20.2 | 11.2 | 8.0 | 89.4 |
| | Friendship | 66.8 | 7.2 | 17.0 | 9.0 | 4.8 | 86.8 |
| | SP | 67.6 | 7.0 | 15.8 | 10.0 | 6.0 | 88.6 |
| | GOT | 67.6 | 5.0 | 15.2 | 9.2 | 6.8 | 88.0 |
| | Social Network | 70.6 | 10.2 | 14.6 | 9.0 | 6.6 | 88.2 |
| | Politician | 71.2 | 8.6 | 16.4 | 9.0 | 6.6 | 87.6 |
| | Expert | 72.4 | 13.6 | 26.2 | 11.2 | 20.8 | 86.4 |
| COT | Overall | 71.9 | 23.8 | 20.7 | 12.6 | 14.0 | 86.7 |
| | Adjacency | 76.0 | 72.4 | 30.2 | 25.6 | 16.6 | 85.4 |
| | Incident | 77.0 | 63.4 | 32.8 | 18.0 | 23.2 | 82.2 |
| | Co-authorship | 67.4 | 22.6 | 19.0 | 13.8 | 9.4 | 84.2 |
| | Friendship | 69.0 | 7.4 | 17.6 | 7.8 | 10.2 | 88.8 |
| | SP | 71.2 | 7.4 | 15.4 | 9.6 | 9.4 | 87.4 |
| | GOT | 71.0 | 9.8 | 16.4 | 7.4 | 10.2 | 89.4 |
| | Social Network | 75.0 | 6.8 | 11.4 | 8.0 | 8.0 | 88.4 |
| | Politician | 72.4 | 10.2 | 15.4 | 11.0 | 13.4 | 89.8 |
| | Expert | 68.2 | 14.6 | 28.4 | 11.8 | 25.2 | 84.8 |
| COT-BAG | Overall | 74.7 | 25.0 | 27.9 | 14.1 | 14.8 | 88.8 |
| | Adjacency | 73.0 | **73.8** | 37.8 | **25.8** | 16.2 | 86.4 |
| | Incident | **80.0** | 63.4 | 35.0 | 19.8 | 23.4 | 84.2 |
| | Co-authorship | 72.4 | 25.2 | 28.8 | 13.8 | 11.2 | 86.0 |
| | Friendship | 74.6 | 8.6 | 25.0 | 11.4 | 8.6 | 90.4 |
| | SP | 74.4 | 9.0 | 23.6 | 10.6 | 9.6 | 90.0 |
| | GOT | 75.6 | 8.0 | 24.8 | 10.2 | 12.4 | **91.8** |
| | Social Network | 76.8 | 9.2 | 19.6 | 9.6 | 12.2 | 90.4 |
| | Politician | 71.2 | 12.6 | 22.2 | 11.8 | 13.4 | **91.8** |
| | Expert | 74.4 | 15.4 | 34.6 | 13.8 | 26.6 | 88.0 |

Table 16: Comparing different graph encoding functions on different graph tasks for PaLM 2 S. The most effective prompting heuristic is highlighted with an underline, and the top-performing graph function encoder for the respective heuristic is highlighted in bold.

| Method | Encoding function | Edge Existence | Node degree | Node count | Edge count | Connected nodes | Cycle check |
|---|---|---|---|---|---|---|---|
| ZERO-SHOT | Overall | 48.2 | 41.4 | 36.5 | 12.1 | 25.0 | 74.7 |
| | Adjacency | 47.2 | 33.6 | 14.0 | 19.0 | 32.8 | **83.6** |
| | Incident | 44.0 | 68.8 | 15.0 | 19.8 | **72.2** | 82.2 |
| | Co-authorship | 49.2 | 36.2 | 41.6 | 12.6 | 14.0 | 57.8 |
| | Friendship | 50.4 | 36.8 | 46.6 | 8.6 | 17.0 | 83.4 |
| | SP | 50.0 | 35.6 | 51.6 | 8.0 | 18.2 | 40.6 |
| | GOT | 51.6 | 36.4 | 49.0 | 11.2 | 15.8 | 75.0 |
| | Social Network | 47.0 | 42.0 | 48.4 | 9.6 | 19.0 | 83.2 |
| | Politician | 49.0 | 41.2 | 37.8 | 7.6 | 14.2 | **83.6** |
| | Expert | 45.8 | 41.8 | 24.8 | 12.4 | 22.0 | 82.6 |
| ZERO-COT | Overall | 37.0 | 7.4 | 31.3 | 13.1 | 14.7 | 16.5 |
| | Adjacency | 46.0 | 36.0 | 20.4 | 20.4 | 15.6 | 69.6 |
| | Incident | 51.2 | 26.8 | 7.8 | 12.0 | 72.4 | 46.6 |
| | Co-authorship | 53.6 | 0.6 | 44.2 | 9.8 | 16.2 | 5.8 |
| | Friendship | 34.6 | 0.4 | 36.4 | 15.8 | 5.2 | 4.8 |
| | SP | 43.8 | 2.6 | 44.6 | 14.8 | 5.6 | 0.6 |
| | GOT | 18.0 | 0 | 43.0 | 12.4 | 2.8 | 0.2 |
| | Social Network | 35.2 | 0 | 38.0 | 22.6 | 2.6 | 3.4 |
| | Politician | 30.4 | 0 | 30.2 | 7.0 | 2.2 | 1.4 |
| | Expert | 20.0 | 0.4 | 17.0 | 3.0 | 9.6 | 16.4 |
| FEW-SHOT | Overall | 47.5 | 36.9 | 40.0 | 16.9 | 22.9 | 40.4 |
| | Adjacency | 61.4 | 37.2 | 81.2 | 18.4 | 37.8 | 43.8 |
| | Incident | 69.4 | 61.2 | 83.2 | 20.4 | 80.4 | 45.8 |
| | Co-authorship | 28.2 | 33.4 | 31.2 | 15.8 | 12.2 | 20.2 |
| | Friendship | 47.2 | 36.0 | 25.6 | 17.4 | 12.8 | 47.2 |
| | SP | 45.6 | 30.6 | 29.4 | 15.8 | 13.8 | 46.4 |
| | GOT | 26.2 | 29.2 | 27.4 | 18.2 | 12.0 | 37.8 |
| | Social Network | 50.4 | 35.8 | 30.8 | 16.6 | 13.8 | 42.4 |
| | Politician | 39.4 | 30.4 | 27.6 | 15.6 | 13.6 | 35.4 |
| | Expert | 60.0 | 38.2 | 23.4 | 14.2 | 10.0 | 45.0 |
| COT | Overall | 57.6 | 44.3 | 41.7 | 19.4 | 23.0 | 42.5 |
| | Adjacency | 62.6 | 69.4 | 82.0 | 23.8 | 41.8 | 41.0 |
| | Incident | **68.2** | **78.4** | 80.8 | 26.6 | 79.6 | 44.4 |
| | Co-authorship | 46.8 | 36.0 | 35.2 | 16.6 | 12.8 | 34.6 |
| | Friendship | 66.2 | 38.0 | 29.0 | 17.4 | 9.6 | 46.4 |
| | SP | 66.4 | 32.4 | 29.2 | 17.6 | 13.8 | 45.6 |
| | GOT | 50.6 | 30.4 | 28.0 | 19.2 | 8.2 | 24.2 |
| | Social Network | 52.2 | 40.4 | 31.4 | 17.0 | 11.0 | 50.4 |
| | Politician | 43.6 | 32.2 | 30.8 | 16.4 | 8.0 | 44.8 |
| | Expert | 62.0 | 41.8 | 28.8 | 20.2 | 22.0 | 51.0 |
| COT-BAG | Overall | 55.2 | 43.7 | 44.4 | 20.8 | 22.7 | 39.6 |
| | Adjacency | 54.0 | 66.2 | 85.2 | **25.8** | 40.6 | 41.6 |
| | Incident | 59.4 | 77.2 | **89.0** | 24.8 | 81.6 | 42.8 |
| | Co-authorship | 41.8 | 38.4 | 35.0 | 18.8 | 11.8 | 29.4 |
| | Friendship | 66.0 | 36.8 | 31.8 | 22.2 | 6.6 | 48.8 |
| | SP | 63.8 | 30.2 | 31.0 | 17.8 | 10.2 | 33.8 |
| | GOT | 59.2 | 32.2 | 31.0 | 19.4 | 10.0 | 21.4 |
| | Social Network | 49.8 | 39.4 | 34.0 | 20.6 | 12.2 | 47.6 |
| | Politician | 47.8 | 32.4 | 32.8 | 16.8 | 6.2 | 38.6 |
| | Expert | 55.4 | 40.6 | 30.0 | 21.0 | 25.4 | 52.8 |

Table 17: Comparing different graph encoding functions on different graph tasks for PaLM 2 L. The most effective prompting heuristic is highlighted with an underline, and the top-performing graph function encoder for the respective heuristic is highlighted in bold.

| Method | Encoding function | Edge Existence | Node degree | Node count | Edge count | Connected nodes | Cycle check |
|---|---|---|---|---|---|---|---|
| ZERO-SHOT | Overall | 47.5 | 55.1 | 76.3 | 30.6 | 19.5 | 83.3 |
| | Adjacency | 43.6 | 49.6 | **100.0** | 36.8 | 55.6 | **83.8** |
| | Incident | 48.6 | 85.0 | 98.6 | 6.6 | 88.0 | 83.2 |
| | Co-authorship | 48.0 | 55.2 | 67.4 | 32.4 | 1.6 | 83.2 |
| | Friendship | 48.0 | 50.8 | 63.2 | 31.2 | 0.2 | 83.2 |
| | SP | 49.2 | 49.8 | 56.4 | 30.8 | 6.6 | 83.2 |
| | GOT | 51.0 | 52.6 | 70.6 | 34.8 | 6.2 | 83.2 |
| | Social Network | 46.0 | 50.2 | 61.0 | 31.6 | 0 | 83.2 |
| | Politician | 50.0 | 52.8 | 70.8 | 32.2 | 1.0 | 83.2 |
| | Expert | 43.0 | 50.2 | 98.6 | 39.4 | 16.0 | 83.2 |
| ZERO-COT | Overall | 41.6 | 7.9 | 73.9 | 24.4 | 39.5 | 22.4 |
| | Adjacency | 31.6 | 13.2 | 66.4 | 25.2 | 61.6 | 52.0 |
| | Incident | 52.0 | 54.8 | 75.0 | 9.8 | 84.4 | 55.2 |
| | Co-authorship | 43.4 | 0.8 | 81.8 | 31.6 | 27.8 | 9.0 |
| | Friendship | 46.0 | 0.6 | 80.2 | 26.2 | 20.4 | 8.4 |
| | SP | 38.6 | 0 | 78.6 | 27.6 | 41.0 | 0.2 |
| | GOT | 38.8 | 0.4 | 68.6 | 30.0 | 29.8 | 1.0 |
| | Social Network | 47.8 | 0 | 78.4 | 30.2 | 28.2 | 6.4 |
| | Politician | 51.4 | 1.2 | 66.6 | 29.2 | 17.2 | 11.6 |
| | Expert | 24.8 | 0.2 | 69.4 | 9.6 | 45.0 | 57.4 |
| FEW-SHOT | Overall | 41.5 | 55.8 | 60.3 | 35.9 | 46.1 | 73.8 |
| | Adjacency | 49.2 | 61.0 | 97.6 | **43.2** | 66.4 | 78.4 |
| | Incident | 73.2 | 82.4 | 99.2 | 37.4 | **85.6** | 78.4 |
| | Co-authorship | 16.4 | 51.4 | 45.4 | 32.2 | 36.8 | 73.6 |
| | Friendship | 32.6 | 53.0 | 45.6 | 35.2 | 44.2 | 79.4 |
| | SP | 33.2 | 45.6 | 40.6 | 35.2 | 30.6 | 57.8 |
| | GOT | 30.0 | 48.6 | 41.6 | 38.2 | 36.0 | 61.4 |
| | Social Network | 40.4 | 51.4 | 44.0 | 32.4 | 38.2 | 79.0 |
| | Politician | 39.4 | 53.6 | 46.2 | 35.2 | 32.2 | 77.6 |
| | Expert | 59.2 | 55.0 | 82.8 | 34.0 | 45.0 | 78.6 |
| COT | Overall | 52.2 | 59.7 | 62.2 | 34.4 | 45.2 | 72.7 |
| | Adjacency | 53.6 | 81.4 | 98.0 | 42.2 | 66.6 | 66.8 |
| | Incident | 72.4 | 94.6 | 98.8 | 29.8 | 87.2 | 68.6 |
| | Co-authorship | 42.4 | 55.0 | 45.0 | 33.8 | 37.0 | 69.4 |
| | Friendship | 55.0 | 48.8 | 45.0 | 33.4 | 40.8 | 80.6 |
| | SP | 54.4 | 48.4 | 44.6 | 34.2 | 26.2 | 71.0 |
| | GOT | 51.4 | 51.2 | 46.2 | 35.2 | 29.4 | 65.4 |
| | Social Network | 42.8 | 51.4 | 43.6 | 32.4 | 39.8 | 77.0 |
| | Politician | 36.8 | 53.0 | 50.4 | 33.0 | 30.2 | 76.8 |
| | Expert | 60.8 | 53.2 | 88.4 | 35.8 | 49.6 | 78.6 |
| COT-BAG | Overall | 60.4 | 60.0 | 63.1 | 34.6 | 45.0 | 69.2 |
| | Adjacency | 64.6 | 78.0 | 98.6 | 39.4 | 64.2 | 70.4 |
| | Incident | **71.6** | **95.4** | 99.4 | 32.2 | 89.0 | 70.8 |
| | Co-authorship | 52.0 | 55.8 | 48.6 | 33.0 | 36.8 | 63.0 |
| | Friendship | 64.8 | 49.6 | 45.6 | 32.6 | 36.4 | 71.8 |
| | SP | 65.8 | 51.6 | 44.2 | 33.0 | 26.4 | 69.4 |
| | GOT | 65.0 | 50.2 | 44.6 | 38.6 | 29.6 | 64.0 |
| | Social Network | 48.4 | 53.6 | 46.4 | 31.2 | 45.6 | 72.0 |
| | Politician | 50.4 | 53.0 | 51.0 | 33.8 | 24.2 | 63.4 |
| | Expert | 61.2 | 52.6 | 89.6 | 37.2 | 52.8 | 78.4 |

