# OpenReview forum: "Talk like a Graph: Encoding Graphs for Large Language Models"
_ICLR.cc/2024/Conference — ICLR 2024 poster_

### Official Review · Reviewer_SLdh · 2023-10-23

**Soundness:** 3 good
**Presentation:** 2 fair
**Contribution:** 3 good
**Rating:** 6
**Confidence:** 4

**Summary:**

This work proposes to understand the graph reasoning abilities of LLMs through a benchmark and experiments. Compared to existing works, this paper uniquely focuses on how to encode graph structures in natural language and different types of graphs, as well as their impact on model performance. Experiments demonstrate that the choice of natural language instantiation and graph structures indeed have an impact on LLMs' ability for graph reasoning.

**Strengths:**

+ reasoning on graphs with LLMs is an important research question
+ the experiments are extensive

**Weaknesses:**

- Since the authors claim the GraphQA benchmark as a novel contribution, it would be great to include at least some description of the benchmark dataset in the main paper. How is the benchmark constructed? What are the hyperparameters in random graph generation? What are the statistics of GraphQA? A brief description of the benchmark in the main paper, accompanied by full details in the appendix, will best help readers understand the scale and validity of the study.

- In equ(2), is it $\max_{g}$ instead of $\max_{g,Q}$?

- It would be nice to have at least a one-sentence description of each graph task in section 3.1. In section 3.5, the *disconnected graph task* is mentioned but it is not introduced at the beginning of section 3.1.

- Since one of the main arguments of this work is "how to encode graphs in natural language affect performance", it would be great to present Table 1 results aggregated by graph encoding functions. It would also be nice to provide hypotheses as to why certain encoding approaches are particularly bad for LLM performance.

- I'm not sure about the uniqueness of some of the findings in this work. Experiments 1-4 in Section 3.1 basically prove two things: 1) LLMs are sensitive to variations in prompt, and 2) larger LMs are generally more capable. While these findings are well established in LLM research, the four experiments simply corroborate them in the graph reasoning domain. I wonder if the authors might have more interpretations of these results beyond those already established in general LLM research.

- For section 4, I wonder if the authors conducted a control experiment, i.e. the only difference among problem subsets is the graph construction algorithm. What factors are specifically fixed in Section 4? It would also be great to provide hypotheses as to why LLMs are better/worse at handling certain graph types.

**Questions:**

please see above

---

> ### Author Response · Authors · 2023-11-16
> **Comment by Authors**
>
> ### Details on GraphQA
>
> Thanks for the suggestion. We added some task descriptions in the main paper (Section 3.1). We also added a separate section for GraphQA in the appendix (A.2) and moved the descriptions under this section and also added more details on it. We are committed to open-source the code and data upon acceptance of the paper.
>
> ### Equation 2
>
> We updated Equation 2 to the following to better reflect the goal of this work:
>
> Our training input $D$ to the graph-based prompt system is a set of ${G, Q, S}$ triples, where $G$ is a graph, $Q$ is a question asked to the LLM, and $S$ is a solution to $Q$, ($S \in W$).  We seek to find a $g(.)$ and $Q$ that maximize the expected score from the model ($\text{score}_f$) of the answers over the training dataset $D$.
>
> \begin{equation}
>   \max_{g, q} E_{G, Q, S \in D} \text{ score}_f(g(G), q(Q), S)
> \end{equation}
>
> Please let us know if this is not clear and we can elaborate more.
>
> ### Adding task descriptions
>
> We added the task descriptions to the beginning of Section 3.1. We also added mode detail for the disconnected node task in Section 3.5.
>
> ### Presenting Table 1 aggregated by graph encoding
>
> Thanks for the great suggestion. We already reported aggregated results in Table 6 in the appendix as the average ranking of each encoder. We also added Table 7 with mean and standard deviation aggregated for each encoder. The results in Table 7 also confirms our hypothesis from Table 6 that the incident encoder outperforms the rest. We posit that the success of incident encoding can be attributed to two key factors. Firstly, it leverages integer node encoding (e.g., *node 0* or *node 1*), as we previously emphasized the advantages of this approach in Section 3.1.1. Secondly, incident edge encoding effectively captures the one-hop neighbourhood around a graph, outperforming methods that simply list edges in a random order.
>
> ### Uniqueness of some of the findings in this work
>
> We agree that many of the findings are in-line with the existing work in the literature. We think that demonstrating the persistence of these (some established) results, in the context of graph reasoning is essential, especially given the widespread adoption of using LLMs for reasoning.
>
> We also showed that linear scaling does not hold for some of the tasks studied here. We observed some non-linear scaling in cycle check (see Figure 3-cycle check) and also in the new results added for the path existence task (checking if a path exists from one node to the other) upon reviewer FDET’s suggestion :
>
> |            | Palm 2 XXS | PaLM 2 XS | PaLM 2 S |
> |------------|------------|-----------|----------|
> | zero-shot  | **80.9**  | 57.5     | **83.7** |
> | zero-cot   | 53.9       | 25.9      | 83.2     |
> | few-shot   | 44.8       | 60.5      | 81.9     |
> | cot        | 53.0       | **63.6**  | 82.4     |
> | cot-bag    | 54.9       | 44.3      | 81.7     |
>
> Another finding of this work was that incident encoding performed the best across all the tasks (discussed in Section A.3).
>
> ### Section 4 experiment setup
>
> The generated graphs for this section are sampled randomly from the generator using similar statistics as input. To be able to report results for this section, we encoded graphs from each graph generator using the nine graph encoding functions and aggregated the results over the graph generator. We believe this setup gives us a controlled experiment to measure the effect of the graph generator.
> We reported some hypotheses in Section 4.2. Here’s a summary:
>
> * For each of the tasks, LLMs have a prior bias about the problem. For instance, for the cycle check task, LLMs have a prior bias saying there is a cycle in the graph. For instance, for the cycle check task, LLMs have a prior bias that assumes there is a cycle in the graph. Therefore, the performance is better for graphs with cycles (e.g., complete graphs) compared to those with no cycles (e.g., star graphs) (Figure 4 visualizes how these graphs differ).
>
> * For some of the tasks with counting (e.g., node degree, node count, and edge count), the fewer the number of edges in the graph, the easier it is for the LLM to reason. For instance, star graphs have the best performance, and complete graphs have the lowest (again, Figure 4 visualizes how these graphs differ).

---

> > ### Comment · Reviewer_SLdh · 2023-11-20
> >
> > I would like to thank the authors for the detailed response and I do not have any outstanding concerns.

---

### Official Review · Reviewer_FDET · 2023-10-29

**Soundness:** 3 good
**Presentation:** 4 excellent
**Contribution:** 2 fair
**Rating:** 6
**Confidence:** 5

**Summary:**

The paper addresses the problem of reasoning on graphs with large language
models (LLMs) and provides a comprehensive exploration of encoding graph-
structured data as text that can use LLMs. The paper claims that the LLM
performance in graph reasoning tasks varies on three crucial fronts: (1) the
method used to encode the graph, (2) the nature of the graph task itself, and
(3) the inherent structure of the graph. The paper has provided comprehensive
experiments on graph reasoning using LLMs by providing them with text prompts
that are constructed from the graphs. In these, the paper analyzes the effect of
a variety of graph-to-text encoding and question encoding functions as well as
graph structures on LLMs performance. Different methods such as Zero-shot,
Few-shot, and Chain-of-Thought methods have been considered for prompting.
To analyze the impact of different graph structures on performance, the paper
has generated random graphs using previous approaches.

**Strengths:**

- The paper has provided detailed discussions of their results along with
reasonable and meaningful conclusions.

- The paper is also well-organized and easy to read.

- The experiments are comprehensive as they include important factors
that can impact the performance of LLMs on graph reasoning. These
are encoding the input graph to text, the structure of the input graph,
rephrasing the question, complexity of the LLM, and prompting method.

**Weaknesses:**

- The graph, node, and edge encoding functions are simple and inefficient.
The paper could use more advanced and recent graph-to-text generation
techniques (i.e. [1]). Evaluating only the defined encoding methods cannot
support the general claims about the power of LLMs in graph reasoning.


- The proposed graph encoding approaches are similar i.e. the Friendship,
Politician, Social network, GOT, and SP all depict alternative ways of
stating two nodes are “connected”. Therefore, evaluating them shows
the power of LLMs in interpreting the names rather than exhibiting their
ability to understand underlying relations and exploit neighborhoods within
a graph. This could have been considered in increasing the diversity of
encoding functions.

- It might be good to introduce previous random graph generation methods.
Adding some detail of these methods (even in the appendix) can be helpful
to understand how they are different.

- The proposed benchmark tasks (except for edge existence) do not involve
reasoning. They can be inferred without reasoning (by counting, simple arithmetic operations, and memorizing the graph structure). More
challenging tasks (e.g., node classification) can enrich the experiments.

- In Experiment 2, authors compare question and application rephrasing
methods, while the difference between these two is not clear. Authors can
add a few examples of rephrasing a question with these methods in the
main body or appendix of their paper.

[1] Yi Luan Mirella Lapata Rik Koncel-Kedziorski, Dhanush Bekal and Han-
naneh Hajishirzi. Text Generation from Knowledge Graphs with Graph
Transformers. In NAACL, 2019.

**Questions:**

It would be great if some of the points raised in the weakness section are addressed.

---

> ### Author Response · Authors · 2023-11-16
> **Comment by Authors**
>
> ### Graph-to-text generation techniques (i.e. [1]) and diversity of the encoding functions
>
> Wow, great suggestion!  In this paper, we specifically studied the class of fixed text encoding functions because we observed that many users of LLM systems are relying on these fixed transformations.  Our results show that your intuition is exactly right (fixed encodings have limitations), and investigating this area is very interesting follow up work!
>
> For this particular reference [1], we searched for an existing model to try to use it “out of the box”, but could not find any available.  Unfortunately we do not have time in the review cycle to appropriately train this as a baseline and collect its results (as running this experiment takes significant time already).
>
> While we initially intended to explore fixed text encoding functions, our goal was to introduce diversity in node and edge encodings. This involved employing a range of node encodings (integers, alphabet letters, popular character names) and edge encodings (friendship, co-authorship, parentheses). By combining these, we achieved a diverse set of graph encoding functions (see Appendix A.1 for more details). The choice of the encoding functions was informed by the typical user practices associated with LLMs.
>
> ### Introduce random graph generation methods in the appendix.
>
> We added descriptions for the random graph generator functions in the appendix in Section A.2.2.
>
> ### More challenging tasks
>
> Thanks for the great suggestion! We have added a synthetic node classification task based on GraphWorld [2], and are currently waiting for the results of this experiment to complete.  (We will update here when this is done)
>
> We would like to note that some of the diverse tasks in our benchmark do share elements of graph classification (especially w.r.t. to motif detection).  For example, The cycle check task needs to do multi-hop reasoning over the structure of the graph in order to determine the existence of a cycle. Also, note that the node degree counting could be viewed as the simplest node regression task.  (More complex versions of this task might try to compute advanced node-level graph properties.)  Similarly, the node and edge count and cycle check might be viewed as  graph classification tasks.
> Upon your suggestion on adding more complex tasks, we conducted experiments on a new task for  path existence which requires multi-hop reasoning. Here are the results:
>
> | Prompting| Palm 2 XXS | PaLM 2 XS | PaLM 2 S |
> |------------|------------|-----------|----------|
> | zero-shot  | **80.9**  | 57.5     | **83.7** |
> | zero-cot   | 53.9      | 25.9     | 83.2     |
> | few-shot   | 44.8      | 60.5     | 81.9     |
> | cot        | 53.0      | **63.6** | 82.4     |
> | cot-bag    | 54.9      | 44.3     | 81.7     |
>
> ### Adding examples for experiment 2.
>
> We added this to the paper. Please let us know if this is not clear. Here’s the examples for your references:
>
> | Task              | Graph question encoder                                  | Application question encoder                       |
> |-------------------|---------------------------------------------------------|-----------------------------------------------------|
> | Edge existence    | Is node Christopher connected to node Michael?          | Are Christopher and Michael friends?                |
> | Node degree       | What is the degree of node 14?                           | How many friends does Christopher have?              |
> | Node count        | How many nodes are in this graph?                         | How many people are mentioned in this information?  |
> | Edge count        | How many edges are in this graph?                         | How many friendships are given in this information?  |
> | Connected nodes   | List all the nodes connected to node Christopher.         | List all the people who are friends with Christopher.|
>
> [1] Yi Luan Mirella Lapata Rik Koncel-Kedziorski, Dhanush Bekal and Han- naneh Hajishirzi. Text Generation from Knowledge Graphs with Graph Transformers. In NAACL, 2019.
>
> [2] Palowitch, John, Anton Tsitsulin, Brandon Mayer, and Bryan Perozzi. "Graphworld: Fake graphs bring real insights for gnns." In Proceedings of the 28th ACM SIGKDD Conference on Knowledge Discovery and Data Mining, pp. 3691-3701. 2022.

---

> > ### Comment · Reviewer_FDET · 2023-11-17
> > **Thanks for the rebuttal!**
> >
> > Thanks for the clarifications and additional results.
> > Based on the rebuttal and the quality of the paper, I am increasing my score.

---

> > > ### Author Response · Authors · 2023-11-17
> > > **Thanks for raising your rating!**
> > >
> > > Thank you for your thoughtful review and for raising your rating. We are grateful for the opportunity to improve our manuscript based on your feedback.

---

> ### Author Response · Authors · 2023-11-20
> **Adding a Node Classification Task**
>
> Your suggestion to explore a node classification task proved to be immensely valuable.
> Inspired by your suggestion, we used our stochastic block model graph generator to create two distinct blocks representing soccer and baseball enthusiasts. We then labeled a subset of nodes and asked the LLM to determine whether an unspecified node belonged to the soccer or baseball group. This exercise aimed to assess the LLM's ability to exploit the homophily in a given graph.
>
> The results for various encoders were as follows:
>
> | Encoding         | Accuracy |
> |------------------|-------|
> | Overall          | 58.4  |
> | Adjacency        | 54.8  |
> | Incident         | 55.2  |
> | Co-authorship    | 68.4  |
> | Friendship       | 74.6  |
> | SP               | 65.8  |
> | GOT              | 51.2  |
> | Social network   | 55.4  |
> | Politician       | 54.2  |
> | Expert           | 46.4  |
>
> The results of this experiment also confirm the major findings in the paper. First, the LLM didn't do well on the node classification task either and only just beat the majority baseline by a small amount (51.2% to 58.4%). Second, the choice of the graph encoder has a significant impact on the LLM reasoning. As this task requires more reasoning (compared to some of the tasks that require more of memorization), some of the encoders with more textual information (e.g., friendship) proved to be more powerful. We can add this experiment to the main paper.
>
> This exercise further demonstrated the versatility of our benchmark, accommodating new tasks and effectively evaluating the performance of LLMs across diverse graph generators. We will open-source the code and data to facilitate future research in this direction.

---

### Official Review · Reviewer_52xH · 2023-10-30

**Soundness:** 3 good
**Presentation:** 3 good
**Contribution:** 2 fair
**Rating:** 6
**Confidence:** 3

**Summary:**

This paper provides an extensive investigation into the capabilities of LLMs in understanding graph structure. The authors explore various factors such as the graph encoding function, prompting questions paradigm, relation encoding, model capacity, and reasoning in the presence of missing edges. The implications of these variables on LLM's graph reasoning and understanding abilities are also carefully examined. Moreover, the authors also investigate the implications of graph structure by randomly generating diverse graphs for evaluation and analyze the results from the impact of graph structure, distractive statements in graph encoding, and the selection of few-shot examples in few-shot learning. This work presents some interesting findings in graph encoding methods, the nature of graph tasks, and the graph structure. The paper yields intriguing findings concerning graph encoding methods, the nature of graph tasks, and the graph structure itself.

**Strengths:**

1. The paper is overall well-written and well-organized, I enjoy reading it.
2. The experiment results are extensive, making it a solid work.
3. I like the analysis in bulletin list style, which helps readers to capture the most important information.
4. There are some interesting findings in this paper.

**Weaknesses:**

1. In the introduction, the authors mention two limitations in the existing LLMs and one of them is difficulty in incorporating fresh information, but how could the graph structure data solve this problem? I would encourage authors to elaborate more on this statement.
2. In section 3.5 experiment 5, the task description is too brief for readers to understand the experimental settings. What is specifically the "disconnected nodes task" and how to generate this data is not clear.
3. The motivation for each experiment setting is not clear enough, I encourage authors to give their motivation in each experiment to help readers understand the necessity for the experiment.
4. For simple tasks such as node degree, node count, edge count, etc. There are some efficient, accurate, and reliable algorithms to do that with programming, so why not just let LLMs write code for these tasks and execute the code to solve these problems?
5. I believe the motivation of this work is not strong enough. Yes, there are graphs everywhere, and reasoning on graphs is essential, but why do we need LLMs to do reasoning on graphs? The LLMs are trained on unstructured textual data, making it hard to generalize to graph data. Moreover, we also have reliable and fast algorithms to solve these basic graph problems, so I believe LLMs might not be a good tool for these basic graph problems.

**Questions:**

N/A

---

> ### Author Response · Authors · 2023-11-16
> **Comment by Authors**
>
> ### 1. How can graphs help LLMs maintain freshness of information?
>
> Graphs are very general data structures -- in this work we are measuring how well structured data can be injected into and interpreted by LLMs.  The connection to freshness goes as follows:
>
> Consider the example of question answering with a LLM which has access to a knowledge graph (KG) or other structured database.
> We can either perform a one-time update [via pre-training or fine-tuning] of the LLMs with the KG information or instead we can provide the KG as in-context information as part of the LLM’s prompt. With the former case, it’s more difficult to update the LLM with fresh information. However, in the latter case, it’s easier to update the database (and the in-context information) as new information becomes available. We updated the second paragraph in the introduction to elaborate on this.
>
> ### 2. Section 3.5 experiment 5: disconnected nodes task
> In this task, we provide a graph description to the LLM, specifying the nodes and edges, and ask about the nodes that are \emph{not} directly connected to a given node. We updated Section 3.5 to better explain this.
>
> ### 3. Adding motivations for each experiment.
> Thanks! We highlighted the motivation for our experiments in the paper.
>
> ### 4 and 5. Motivation for using LLMs for graph tasks.
>
> Great question!  Our motivation is not to suggest that LLMs replace the reliable and efficient graph algorithms we all grew up with -- instead it's quite the opposite.  We believe that graph tasks are an ideal lens for understanding multi-step reasoning with LLMs.  By quantifying LLMs (poor) performance on these tasks, we highlight a significant gap with current technologies and open the door for future research in the area.
>
> This is especially interesting, because many tasks that people ask of LLMs today reduce to graph algorithms (or have graph algorithms as essential components of their solution).  For instance, when using LLMs in problems requiring deductive logical reasoning, the LLM should be able to perform edge existence, and reason about connected/disconnected nodes.  However, there is not much work actually evaluating these capabilities.  This is especially important for cases when the question is textual (and therefore common sense knowledge is needed) and it is not obvious what out-of-the-box graph algorithm might even apply.

---

> > ### Author Response · Authors · 2023-11-20
> > **Discussion period ending soon**
> >
> > Dear reviewer,
> >
> > We are writing to kindly remind you that the discussion period is coming to an end soon. We believe our paper is standing much stronger after incorporating your constructive feedback, and we would really appreciate it if you could take some time to read our responses and provide any further feedback that you have. Your feedback is important to us, and we would like to have the opportunity to discuss them with you before the discussion period ends. Thank you again for helping us, and the remainder of the academic community.

---

### Official Review · Reviewer_PSY3 · 2023-11-01

**Soundness:** 3 good
**Presentation:** 2 fair
**Contribution:** 2 fair
**Rating:** 6
**Confidence:** 3

**Summary:**

This work presents the first comprehensive study on encoding graph-structured data as text for large language models (LLMs). Graphs are widely used to represent complex relationships in various applications, and reasoning on graphs is crucial for uncovering patterns and trends. The study reveals that LLM performance in graph reasoning tasks depends on three key factors: the graph encoding method, the nature of the graph task itself, and the structure of the graph considered. These findings provide valuable insights into strategies for improving LLM performance on graph reasoning tasks, with potential performance boosts ranging from 4.8% to 61.8%, depending on the specific task.

**Strengths:**

* It is a valuable problem for encoding graph-structured data as text for LLMs.
* Many factors are taken into considerations, and detailed analyses are provided.
* The findings provide valuable insights into strategies for improving LLM performance on graph reasoning tasks.

**Weaknesses:**

1. One concern is about the experiment. The paper explores encoding graph-structured data as text for **LLMs**. However, only one type of LLM is compared (PaLM). It would be better to make comparisons with other LLMs, like GPT3/4 and Llama to make the findings more convincing.

2. Another concern is about the novelty. The proposed graph encoder function g() in this paper is a mapping from graph space to textual space. Several previous paper [1-3] explores describing graph neighbors in natural language, and it would be better to tell the difference of this work.

[1] Guo, Jiayan, Lun Du, and Hengyu Liu. "GPT4Graph: Can Large Language Models Understand Graph Structured Data? An Empirical Evaluation and Benchmarking." arXiv preprint arXiv:2305.15066 (2023).

[2] Chen, Zhikai, et al. "Exploring the potential of large language models (llms) in learning on graphs." arXiv preprint arXiv:2307.03393 (2023).

[3] Ye, Ruosong, et al. "Natural language is all a graph needs." arXiv preprint arXiv:2308.07134 (2023).

**Questions:**

See Weaknesses.

---

> ### Author Response · Authors · 2023-11-16
> **Comment by Authors**
>
> ### comparisons with other LLMs
>
> We appreciate the reviewer's valuable suggestion to conduct experiments on other LLMs. We run our experiments using GPT3.5. This additional evaluation provides further insights into the performance of our approach and confirms the key findings observed on Palm 1 and 2.
>
> Graph encoder functions have a substantial impact on the performance of LLMs on graph-based tasks, with the node degree task exhibiting a performance variance of 5.4% to 66.4%. Integer node encoding enhances arithmetic performance for node degree, node count, and edge count tasks. The incident encoding outperforms other encoding functions by a significant margin on the connected node task due to its ability to make information more readily accessible for LLM utilization.
>
>
>
> | Encoding function | Edge Existence | Node degree | Node count | Edge count | Connected nodes | Cycle check |
> |-------------------|----------------|-------------|------------|------------|------------------|-------------|
> | Overall           | 77.0           | 42.2        | 98.8       | 37.6       | 46.7             | 86.1        |
> | Adjacency         | 73.0           | 5.4         | 99.2       | **46.2**   | 59.2             | 84.2        |
> | Incident          | 80.2           | **66.4**    | 96.2       | 12.6       | **75.2**         | 84.6        |
> | Co-authorship     | **82.0**       | 42.6        | 99.8       | 39.4       | 37.2             | 86.4        |
> | Friendship        | 73.8           | 43.8        | **100.0**  | 41.8       | 41.2             | 87.4        |
> | SP                | 74.0           | 44.2        | 99.8       | 39.0       | 38.6             | 86.4        |
> | GOT               | 75.8           | 41.0        | 99.0       | 39.4       | 38.6             | **88.6**    |
> | Social Network    | 78.6           | 47.6        | 99.8       | 38.6       | 40.6             | 86.6        |
> | Politician        | 78.6           | 45.4        | **100.0**  | 39.8       | 40.2             | 84.2        |
> | Expert            | 75.7           | 43.5        | 95.3       | 41.7       | 49.2             | 86.6        |
>
> ### Novelty wrt [1, 2, 3]:
>
> Thanks for the great question. Our work uniquely investigates how properties of the graph structure (via synthetic generation) influence the choice of graph encoding function. This is completely novel, as no other related work examines these interactions. This novel approach enables us to systematically study these interactions, which have not been explored in previous research. Additionally, we conducted extensive scaling experiments with LLMs of varying parameter sizes which no other work has done. Finally, we have significant differences from each work individually, for example:  unlike [1], we study a diversity of LLMs (and have added more models in the rebuttal), unlike [2] we vary prompting and graph tasks, and unlike [3], we use a black box model without model weights.. Furthermore, we conducted experiments on varying question encoder functions to evaluate their impact on performance.
>
> In order to clearly state the novelty of our work, we have added this following comparison table to the paper:
>
> |                               | Guo et al.[1] | Chen et al. [2] | ye et al. [3] | Ours         |
> |-------------------------------|-------------------|--------------------------|-----------------------|--------------|
> | Synthetic generation          | &#10005;          | &#10005;                 | &#10005;              | &#10004;   |
> | Black-box model               | &#10004;        | &#10004;               | &#10005;              | &#10004;   |
> | Scaling experiments           | &#10005;          | &#10005;                 | &#10005;              | &#10004;   |
> | Varying question encoding function | &#10005;     | &#10005;                | &#10005;              | &#10004;   |
> | Varying edge encoding function | &#10005;          | &#10005;                 | &#10005;              | &#10004;   |
> | Varying graph structure        | &#10005;          | &#10005;                 | &#10005;              | &#10004;   |
> | Varying graph encoding function | &#10004;        | &#10005;                 | &#10005;              | &#10004;   |

---

> > ### Author Response · Authors · 2023-11-20
> > **Discussion period ending soon**
> >
> > Dear reviewer,
> >
> > We are writing to kindly remind you that the discussion period is coming to an end soon. We believe our paper is standing much stronger after incorporating your constructive feedback, and we would really appreciate it if you could take some time to read our responses and provide any further feedback that you have. Your feedback is important to us, and we would like to have the opportunity to discuss them with you before the discussion period ends. Thank you again for helping us, and the remainder of the academic community.

---

### Author Response · Authors · 2023-11-16
**Response to all**

We would like to thank all the reviewers for taking a considerable amount of their busy schedules to give us valuable and constructive feedback. We took all of your comments, and our paper is standing much stronger. Thank you for helping us, and the remainder of the academic community.

We respond to each reviewer, individually.

---

### Meta-Review · Area_Chair_JBgi · 2023-12-15

**Metareview:**

This paper conducts an empirical study on prompting graph data into LLMs to solve graph learning tasks. The study is comprehensive, covering a variety of datasets and tasks. There have been several studies exploring the same topic so claiming as the "first comprehensive" study of this type is a bit overclaiming. This is certainly not the first study. While this study explores certain novel aspects that were not covered by existing studies, when can we call a study "comprehensive" is very subjective. With that said, the inclusion of the synthetic graph generation does provide interesting results, which might be of interest to the community.

**Justification For Why Not Higher Score:**

This study is not entirely novel.

**Justification For Why Not Lower Score:**

This study provides certain novel aspects not covered by existing ones.

---

### Decision · Program_Chairs · 2024-01-16

Accept (poster)